# CAMKII and Calcineurin regulate the lifespan of *Caenorhabditis elegans* through the FOXO transcription factor DAF-16

**Li Tao[1,2], Qi Xie[3], Yue-He Ding[1,2], Shang-Tong Li[2], Shengyi Peng[3], Yan-Ping Zhang[2], Dan Tan[2], Zengqiang Yuan[3]\*, Meng-Qiu Dong[1,2]\***

[1]Graduate Program in Chinese Academy of Medical Sciences and Peking Union Medical College, Beijing, China; [2]National Institute of Biological Sciences, Beijing, Beijing, China; [3]Institute of Biophysics, Chinese Academy of Sciences, Beijing, China

**Abstract** The insulin-like signaling pathway maintains a relatively short wild-type lifespan in *Caenorhabditis elegans* by phosphorylating and inactivating DAF-16, the ortholog of the FOXO transcription factors of mammalian cells. DAF-16 is phosphorylated by the AKT kinases, preventing its nuclear translocation. Calcineurin (PP2B phosphatase) also limits the lifespan of *C. elegans*, but the mechanism through which it does so is unknown. Herein, we show that TAX-6•CNB-1 and UNC-43, the *C. elegans* Calcineurin and Ca$^{2+}$/calmodulin-dependent kinase type II (CAMKII) orthologs, respectively, also regulate lifespan through DAF-16. Moreover, UNC-43 regulates DAF-16 in response to various stress conditions, including starvation, heat or oxidative stress, and cooperatively contributes to lifespan regulation by insulin signaling. However, unlike insulin signaling, UNC-43 phosphorylates and activates DAF-16, thus promoting its nuclear localization. The phosphorylation of DAF-16 at S286 by UNC-43 is removed by TAX-6•CNB-1, leading to DAF-16 inactivation. Mammalian FOXO3 is also regulated by CAMKIIA and Calcineurin.

**\*For correspondence:** zqyuan@
sun5.ibp.ac.cn (ZY);
dongmengqiu@nibs.ac.cn
(M-QD)

**Competing interests:** The authors declare that no competing interests exist.

**Reviewing editor**: Michael Czech, University of Massachusetts Medical School, United States

## Introduction

Multiple signaling pathways, including insulin/IGF-1 signaling (IIS), germline signaling, mitochondrial signaling, and the signaling pathways induced by dietary restriction, regulate aging in *C. elegans* (*Kenyon, 2005*; *Greer and Brunet, 2008*). The best characterized is the IIS pathway, which includes the upstream insulin/IGF-1 receptor DAF-2 and the downstream FOXO transcription factor DAF-16. Signals from DAF-2 are transmitted through AGE-1 (phosphoinositide 3-kinase) and PDK-1 (phosphoinositol-dependent kinase-1) to AKT-1, AKT-2, and SGK-1, which phosphorylate DAF-16 and prevent it from translocating to the nucleus to activate a pro-longevity gene network. Reduction-of-function mutations of the kinase genes in the IIS pathway, from *daf-2* to *akt-1, akt-2,* and *sgk-1*, all extend lifespan in a *daf-16*-dependent manner (*Kenyon et al., 1993*; *Morris et al., 1996*; *Kimura et al., 1997*; *Hertweck et al., 2004*). Lifespan extension by reduced IIS is also observed in other species, including flies and mice, indicating that the pathway is conserved evolutionarily (*Kenyon, 2005*; *Greer and Brunet, 2008*).

Compared with the kinases in the *C. elegans* IIS pathway, little is known about the protein phosphatases that neutralize the effects of the kinases. The only known example is PPTR-1, a B56 regulatory subunit of PP2A, which directs PP2A to dephosphorylate AKT-1 at T350, thereby inactivating the kinase (*Padmanabhan et al., 2009*). DAF-18, the *C. elegans* PTEN, is a phosphatidylinositol 3,4,5-trisphosphate (PIP3) 3-phosphatase (*Ogg and Ruvkun, 1998*). The phosphatase for DAF-2 has not been identified, nor the one that regulates DAF-16. Previously, *loss-of-function (lf)* mutants for *tax-6* and *cnb-1*, which encode the catalytic and regulatory subunits of *C. elegans* Calcineurin,

**eLife digest** Although aging might seem to be a passive process—resulting simply from wear and tear over a lifetime—it can actually be accelerated or slowed down by genetic mutations. This phenomenon has been most thoroughly studied in the nematode worm, *Caenorhabditis elegans*. Normally, this worm lives for just two or three weeks, but genetic mutations that reduce the activity of certain enzymes in a series of biochemical reactions known as the insulin/IGF-1 signalling pathway can extend its lifespan by up to a factor of ten, and similar effects have been seen in flies and mice. Lifespans can also be increased by blocking other signalling pathways or restricting the intake of calories.

This increase in lifespan associated with the insulin/IGF-1 signalling pathway is known to involve a protein called DAF-16 and two kinases called AKT-1 and AKT-2. Under normal conditions the AKT kinases add several phosphate groups to the DAF-16, which prevents it from travelling to the nucleus of the cell. However, when genetic techniques are used to block the insulin/IGF-1 signalling pathway, the AKT kinases are unable to add the phosphate groups; this leaves the DAF-16 free to enter the nucleus, where it activates a network of genes that promotes longevity.

In addition to kinases, the insulin/IGF-1 signalling pathway also involves enzymes called phosphatases that remove the phosphate groups from other proteins. In particular, a phosphatase called calcineurin is known to be involved in the regulation of lifespan, but the details of this process are not fully understood.

Now, Tao et al. have carried out a series of genetic and biochemical experiments to determine how phosphatases exert their influence on aging. The results show that calcineurin targets DAF-16, the same protein that is targeted by the AKT kinases. Moreover, another kinase also targets DAF-16 when the worm is exposed to heat, starvation or some other form of stress: this kinase, which is not involved in the insulin/IGF-1 signalling pathway, is called CAMKII.

Tao et al. show that these kinases act on DAF-16 in different ways: CAMKII activates it by adding the phosphate group at a specific site known as S286, whereas the AKT kinases deactivate DAF-16 because they add phosphate groups at different sites, thereby preventing it from entering the nucleus. Calcineurin neutralizes the effect of CAMKII by removing the phosphate group at S286 to deactivate the DAF-16.

In addition to shedding new light on the regulation of lifespan in *C. elegans*, the new results could improve our understanding of aging in humans, and also the development of diabetes and other age-related diseases, because the equivalent molecules in mammalian cells are regulated in similar ways.

---

respectively, were found to live longer than wild-type (WT) worms (*Dong et al., 2007*). In mammalian systems, Calcineurin (PP2B) is a $Ca^{2+}$/calmodulin-dependent serine/threonine protein phosphatase that has diverse functions and affects both T cell activation and heart development (*Crabtree, 1999*). In *C. elegans*, Calcineurin regulates body size, thermotaxis, muscle contraction, and lifespan (*Bandyopadhyay et al., 2002*; *Kuhara et al., 2002*; *Lee et al., 2004*; *Dong et al., 2007*). The longevity phenotype of *tax-6(lf)* is partially dependent on *daf-16* (*Dong et al., 2007*). More recent studies have shown that *C. elegans* Calcineurin can regulate lifespan by suppressing autophagy (*Dwivedi et al., 2009*) or inactivating CRTC-1, a co-activator of CREB (*Mair et al., 2011*). However, direct targets of worm Calcineurin have not been identified.

In the current work, we addressed how worm Calcineurin TAX-6•CNB-1 regulates lifespan. We discovered that DAF-16 was phosphorylated and activated by UNC-43 at the serine 286 (S286) site. The phosphoryl group was removed by TAX-6•CNB-1. UNC-43 and TAX-6•CNB-1 therefore regulate *C. elegans* lifespan through the reversible phosphorylation of DAF-16. This regulatory mechanism has a different mode of action from the canonical IIS pathway because the phosphorylation activates, rather than represses, DAF-16. Activation of DAF-16 by UNC-43 occurs in response to different types of stress signals, such as heat, starvation, and oxidation. UNC-43 and TAX-6•CNB-1 can regulate DAF-16 independently of IIS, and the two signaling mechanisms appear to crosstalk, leading to coordinated action on DAF-16. We also show that the regulation of FOXO by CAMKII and Calcineurin is conserved in mammalian cells.

## Results

### TAX-6 interacts with DAF-16 in vitro and in vivo

A genetic analysis has shown that a *daf-16(null)* allele partially suppresses the longevity of the *tax-6(lf)* mutant (*Dong et al., 2007*). This observation suggested that *daf-16* is either a direct or indirect downstream target of *C. elegans* Calcineurin; alternatively, it acts independently. To sort through these possibilities, we immunoprecipitated the 3xFLAG::DAF-16 protein using an anti-FLAG antibody from the lysate of MQD82, a transgenic worm strain that expresses this fusion protein and TAX-6::GFP (*Figure 1A*, lane 1). TAX-6::GFP from this lysate was co-precipitated with the FLAG antibody (*Figure 1A*, lane 4), but no TAX-6::GFP was co-precipitated from the lysate of MQD2, a strain expressing only TAX-6::GFP (*Figure 1A*, lanes 2 and 5). From the mixed lysates of two other transgenic strains, one expressing 3xFLAG::DAF-16 and the other expressing GFP, the GFP protein also failed to be precipitated by the FLAG antibody (*Figure 1A*, lanes 3 and 6). This result suggested that TAX-6 and DAF-16 interacted with each other in vivo. To determine whether the interaction was direct, we individually purified the

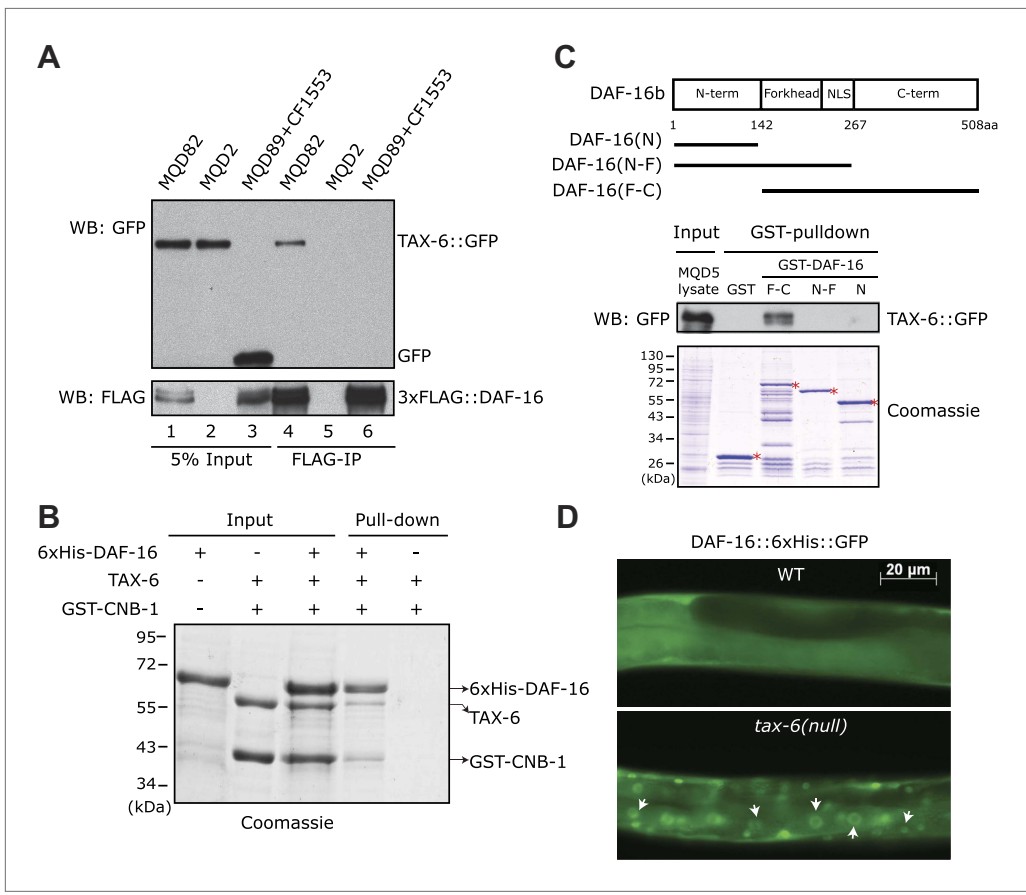

**Figure 1**. *C. elegans* Calcineurin TAX-6•CNB-1 directly binds to DAF-16 and negatively regulates DAF-16 nuclear localization. (**A**) DAF-16 and TAX-6 form a complex in vivo. Immunoprecipitation of 3xFLAG::DAF-16 expressed under the *daf-16* promoter in WT *C. elegans* pulled down TAX-6::GFP expressed under the *tax-6* promoter. The lysates were obtained from transgenic strains MQD82 (co-expressing 3xFLAG::DAF-16 and TAX-6::GFP), MQD2 (expressing TAX-6::GFP), MQD89 (expressing 3xFLAG::DAF-16), and CF1553 (expressing GFP under a *sod-3* promoter). (**B**) Calcineurin directly binds to DAF-16. Purified recombinant TAX-6•GST-CNB-1 was pulled down with Ni-NTA beads through its interaction with purified His-tagged DAF-16. (**C**) The C-terminal region of DAF-16 most likely mediates the interaction with Calcineurin. GST-DAF-16(F-C), but not GST, GST-DAF-16(N) or GST-DAF-16(N-F), pulled down TAX-6::GFP expressed in *C. elegans* (strain MQD5). The DAF-16 C-terminal region alone was not stable. Asterisk indicates full-length GST or GST fusion proteins. (**D**) DAF-16::6xHis::GFP is diffusely distributed in the WT animals but concentrated in the nucleus in *tax-6(ok2065)* animals. All GFP images shown in this paper are of L4 animals at 20°C unless otherwise indicated.

recombinant His-tagged DAF-16 and *C. elegans* Calcineurin in the form of TAX-6•GST-CNB-1 and then mixed them together before pulling down DAF-16 with nickel beads. Indeed, TAX-6•GST-CNB-1 was pulled down successfully using His-tagged DAF-16 (*Figure 1B*). Further analysis suggested that the C-terminal region of DAF-16 mediates the interaction with Calcineurin (*Figure 1C*). Together, these results raised the interesting possibility that Calcineurin might directly regulate DAF-16 in *C. elegans*.

## TAX-6 negatively regulates DAF-16 nuclear translocation

In WT animals, DAF-16 is phosphorylated by AKT and diffusely distributed throughout the cell, whereas in long-lived IIS mutants, transcriptionally active DAF-16 accumulates in the nucleus (*Henderson and Johnson, 2001*; *Lin et al., 2001*). To determine whether DAF-16 translocates to the nucleus in long-lived Calcineurin mutants, we expressed DAF-16::6xHis::GFP using a *daf-16* promoter in the *tax-6(ok2065)* background. The functionality of this transgene was verified in *daf-2(e1370ts);daf-16(mu86)* animals; the double mutant expressing this transgene displayed the dauer formation-constitutive (Daf-c) and longevity phenotypes of the *daf-2(e1370ts)* single mutant (not shown). DAF-16::6xHis::GFP clearly accumulated in the nuclei of different types of cells in *tax-6(ok2065)* (*Figure 1D*), similar to *daf-2(lf)* and *akt-1;akt-2(RNAi)* mutants (*Henderson and Johnson, 2001*).

The above results strongly suggested that Calcineurin directly inhibited DAF-16, presumably by dephosphorylating it. This result is in contrast to the transmission of IIS signaling to DAF-16, during which phosphorylation of DAF-16 by AKT inhibits its nuclear accumulation. Thus, AKT and Calcineurin regulate the phosphorylation of different sites on DAF-16; otherwise, *tax-6(lf)* would have a lifespan phenotype opposite to that of *akt-1;akt-2(lf)* or *daf-2(lf)*. In contrast, they all live longer than WT worms. Therefore, Calcineurin must counteract a different kinase that activates DAF-16.

## UNC-43, the *C. elegans* CAMKII homolog, promotes DAF-16 nuclear translocation and longevity

We reasoned that nuclear accumulation of DAF-16 in the *tax-6(null)* background should be dependent on a kinase that initiated the phosphorylation and nuclear translocation of DAF-16. Inactivation of this kinase should abolish the nuclear accumulation of DAF-16::GFP in *tax-6(null)* animals. We thus conducted an RNAi screen of kinase genes using the DAF-16::6xHis::GFP reporter in *tax-6(ok2065)* worms, in which the GFP signal accumulates in the nucleus (*Figure 2—figure supplement 1*). We obtained from the Ahringer library a strong suppressor of nuclear DAF-16::GFP in an RNAi clone that targets *unc-43* (*Figure 2—figure supplement 2*). The suppression was confirmed by an independent, homemade *unc-43* RNAi construct (*Figure 2A*). The *unc-43* gene encodes the only Ca$^{2+}$/calmodulin-dependent serine/threonine protein kinase, type II (CAMKII), in *C. elegans* (*Reiner et al., 1999*). UNC-43 and Calcineurin have opposing functions in locomotion and egg laying, two behavioral phenotypes that are regulated by G protein signaling (*Reiner et al., 1999*; *Bandyopadhyay et al., 2002*). However, they also have non-overlapping functions because *unc-43(lf);cnb-1(null)* double mutant animals arrest at the L1 larval stage, whereas both single mutants are viable (*Reiner et al., 1999*; *Bandyopadhyay et al., 2002*).

To verify that UNC-43 and Calcineurin regulate DAF-16 antagonistically, we examined whether *unc-43(RNAi)* could suppress the longevity of Calcineurin loss-of-function mutants and whether *unc-43(gf)* had the same phenotype as *tax-6(lf)*. Although RNAi knockdown of *tax-6* during adulthood increased the lifespan of WT worms (*Figure 2B*, *Figure 2—source data 1A*), it failed to increase the lifespan of *unc-43(null)* animals (*Figure 2C*, *Figure 2—source data 1A*). This observation is consistent with the hypothesis that Calcineurin removes the activating phosphorylation on DAF-16 caused by UNC-43. We then examined *n498*, an *unc-43 gain-of-function (gf)* allele harboring a missense mutation that results in constitutive activation of the kinase (*Park and Horvitz, 1986*; *Reiner et al., 1999*). This *unc-43(gf)* allele caused DAF-16::6xHis::GFP to accumulate in the nucleus (*Figure 2D*), phenocopying *tax-6(lf)* (*Figure 2A*). Furthermore, the *unc-43(n498)* mutant lived 80% longer than the WT animals, and this lifespan extension was largely dependent on *daf-16* (*Figure 2E*, *Figure 2—source data 1B*).

Consistent with UNC-43 activating DAF-16 to prolong lifespan and TAX-6•CNB-1 antagonizing such action, we found that the *unc-43(gf);cnb-1(lf)* double mutant lived even longer than either single mutant (*Figure 2F*, *Figure 2—source data 1C*).

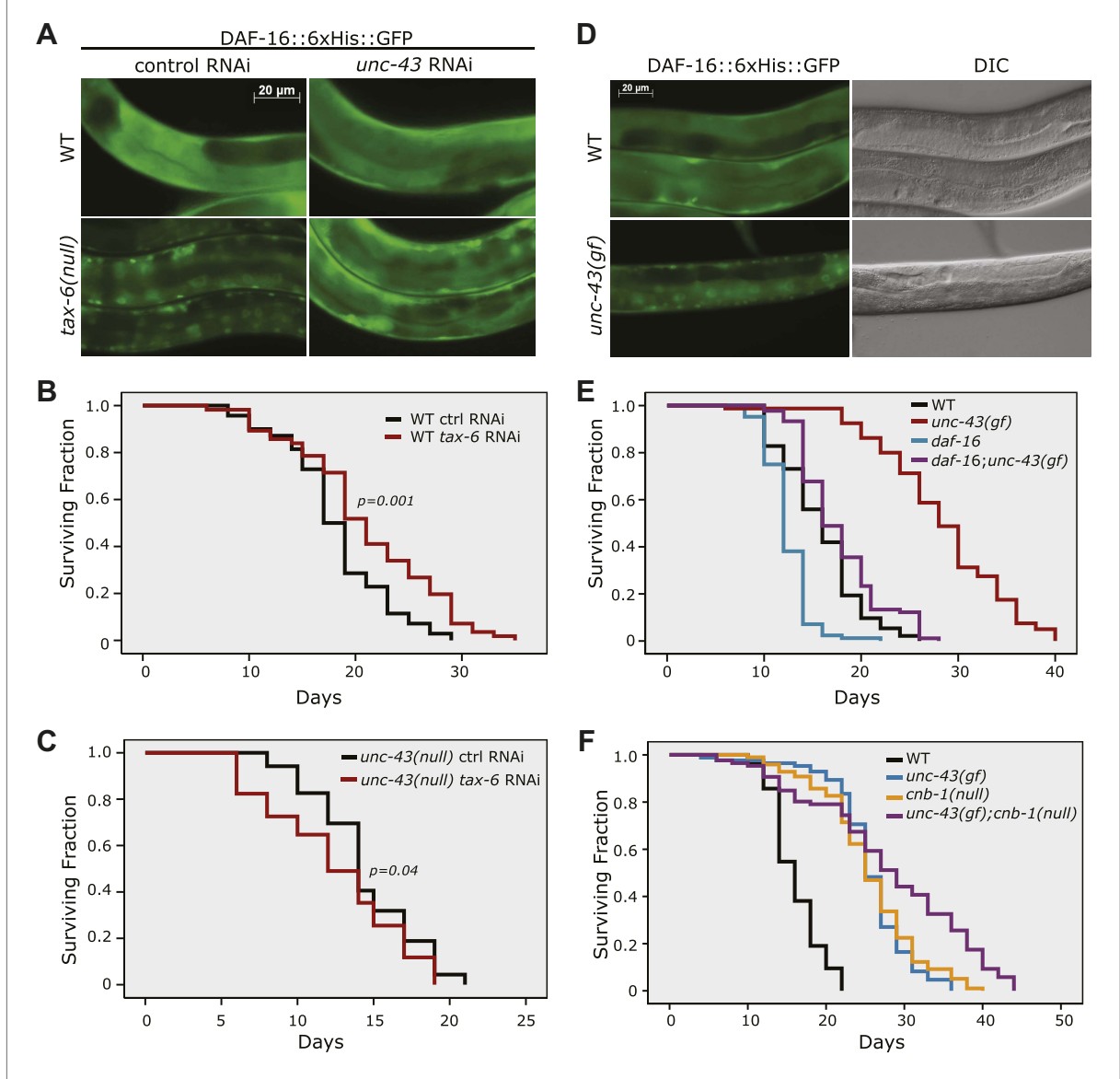

**Figure 2**. The effect of *tax-6(lf)* mutations on DAF-16 localization and lifespan requires *unc-43,* the CAMKII gene whose effect opposes that of *tax-6*. (**A**) *unc-43* RNAi abolished DAF-16 nuclear accumulation in *tax-6(ok2065)*. Adult stage RNAi knockdown of *tax-6* extended the WT lifespan (p=0.001) (**B**) but may have slightly shortened the lifespan of *unc-43(n498n1186)*, a putative null mutant (p=0.04) (**C**). A constitutively active gain-of-function mutation, *unc-43(n498)*, caused DAF-16::6xHis::GFP to accumulate in the nucleus (**D**) and extended lifespan in a largely *daf-16*-dependent manner (**E**). p<0.001 for *daf-16(mu86);unc-43(gf)* vs *daf-16(mu86)* or *unc-43(gf)*. The log-rank p values are reported for all lifespan data in this study. (**F**) The *unc-43(gf);cnb-1(null)* double mutant has a longer lifespan than both the *unc-43(gf)* and *cnb-1(null)* mutants, and all three mutant strains live longer than WT animals. p<0.001 for WT vs any mutant, and p<0.001 for *unc-43(gf);cnb-1(null)* vs *unc-43(gf)* and *cnb-1(null)*.

The following source data and figure supplements are available for figure 2:

**Source data 1**. The *unc-43(null)* mutant showed a WT-like lifespan that was epistatic to the longevity effect of *tax-6(RNAi)*, while the *unc-43(gf)* mutant was long-lived.

**Figure supplement 1**. A screen for the kinase(s) required for the nuclear accumulation of DAF-16::GFP induced by *tax-6(null)*.

**Figure supplement 2**. An *unc-43* RNAi clone from the Ahringer library suppressed the nuclear accumulation of DAF-16::GFP induced by *tax-6(null)*.

# UNC-43 activates DAF-16 independent of the NSY-1/SEK-1 MAP kinase pathway

Previous studies have shown that UNC-43 activates NSY-1 and SEK-1, two kinases within the MAP kinase pathway, to repress *str-2* expression (*Sagasti et al., 2001*). To determine whether UNC-43 activated DAF-16 through NSY-1 (MAPKKK) and SEK-1 (MAPKK), we treated *unc-43(n498)* worms expressing the DAF-16::6xHis::GFP transgene with *nsy-1* and *sek-1* RNAi and found that neither gene affected DAF-16 nuclear accumulation induced by *unc-43(gf)* (*Figure 3A*). RNAi against *pmk-2*, which

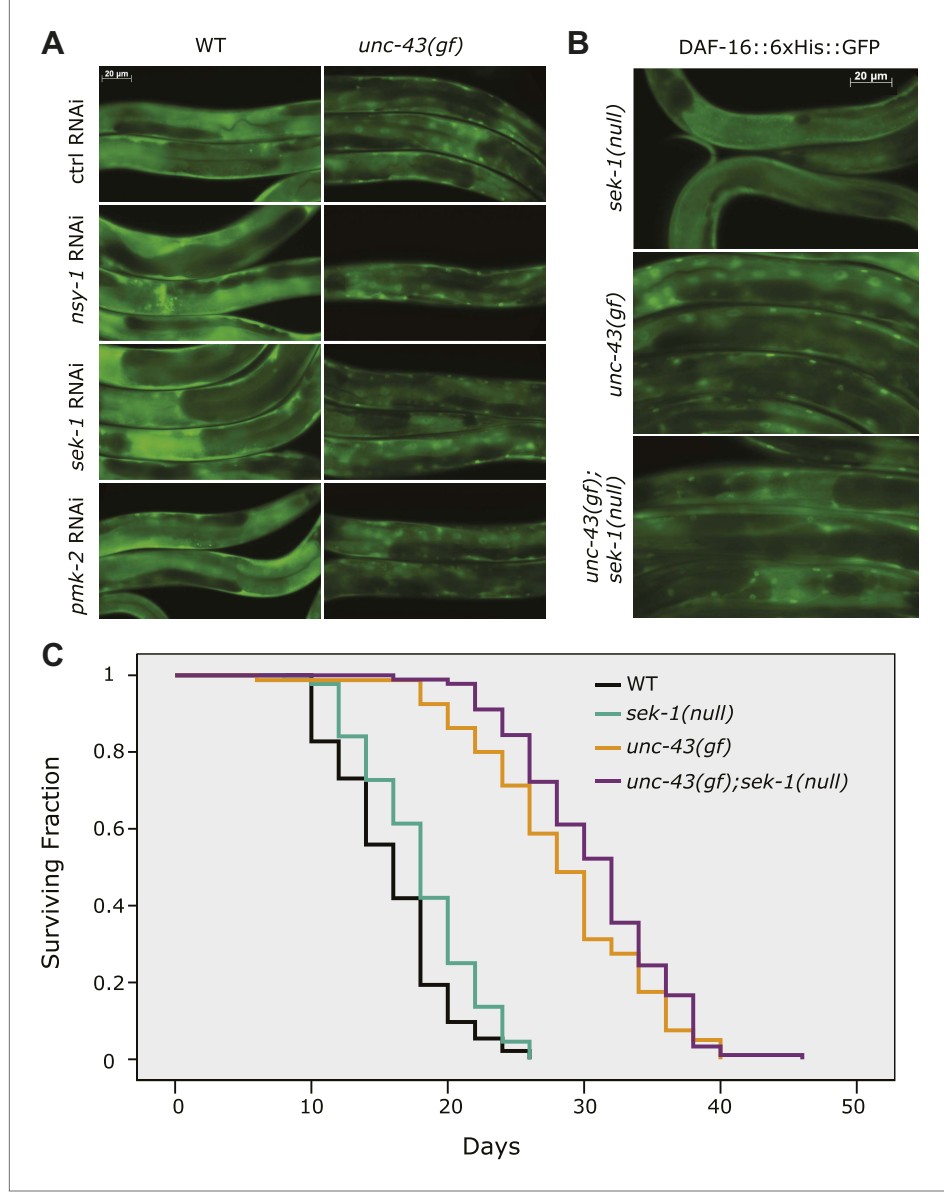

**Figure 3**. UNC-43 does not regulate DAF-16 localization through the NSY-1/SEK-1 MAPK kinase pathway. (**A**) RNAi knockdown of *nsy-1*, *sek-1*, and *pmk-2* failed to block the nuclear accumulation of DAF-16::6xHis::GFP in *unc-43(gf)* mutants. Worms were fed with the indicated RNAi bacteria from hatching and imaged at the L4 stage. (**B**) Similar to *sek-1* RNAi, the *sek-1(km4)* mutation did not eliminate DAF-16 nuclear localization in *unc-43(gf)* worms. (**C**) *km4*, a null allele of *sek-1*, did not shorten the lifespan of WT or *unc-43(gf)* animals. In contrast, *km4* may have slightly extended their lifespan. p=0.001 for *sek-1(null)* vs WT and p=0.047 for *unc-43(gf); sek-1(null)* vs *unc-43(gf)*.

The following source data are available for figure 3:

**Source data 1**. *km4*, the null allele of *sek-1*, did not shorten the lifespan of *unc-43(gf)* worms.

encodes a p38 MAP kinase (*Kim et al., 2002*; *Tanaka-Hino et al., 2002*), also had no effect (*Figure 3A*). *sek-1* regulates DAF-16 localization in response to oxidative stress (*Kondo et al., 2005*). To ascertain the role of *sek-1* in the nuclear accumulation of DAF-16 induced by *unc-43(gf)*, we crossed the DAF-16::6xHis::GFP animal to a *unc-43(gf);sek-1(null)* double mutant. Similar to the RNAi treatment, the *km4(null)* allele of *sek-1* failed to prevent DAF-16 nuclear localization in *unc-43(gf)* animals (*Figure 3B*). Moreover, the *unc-43(gf);sek-1(null)* double mutant did not live a shorter life than *unc-43(gf)* animal (*Figure 3C*, *Figure 3—source data 1*). Thus, we conclude that UNC-43 does not activate DAF-16 through the NSY-1/SEK-1 MAP kinase pathway.

## UNC-43 can directly bind to and phosphorylate DAF-16

To test whether DAF-16 is a direct substrate of UNC-43, we constructed transgenic strains that co-expressed 3xHA::UNC-43 and 3xFLAG::DAF-16 (MQD522) or expressed either 3xHA::UNC-43 (MQD530) or 3xFLAG::DAF-16 (MQD89) under the control of own promoter. The 3xHA::UNC-43 protein was found in the immunoprecipitate of 3xFLAG::DAF-16 and vice versa (*Figure 4A–B*), suggesting that UNC-43 and DAF-16 can form a complex in vivo. Using purified recombinant proteins, we found that

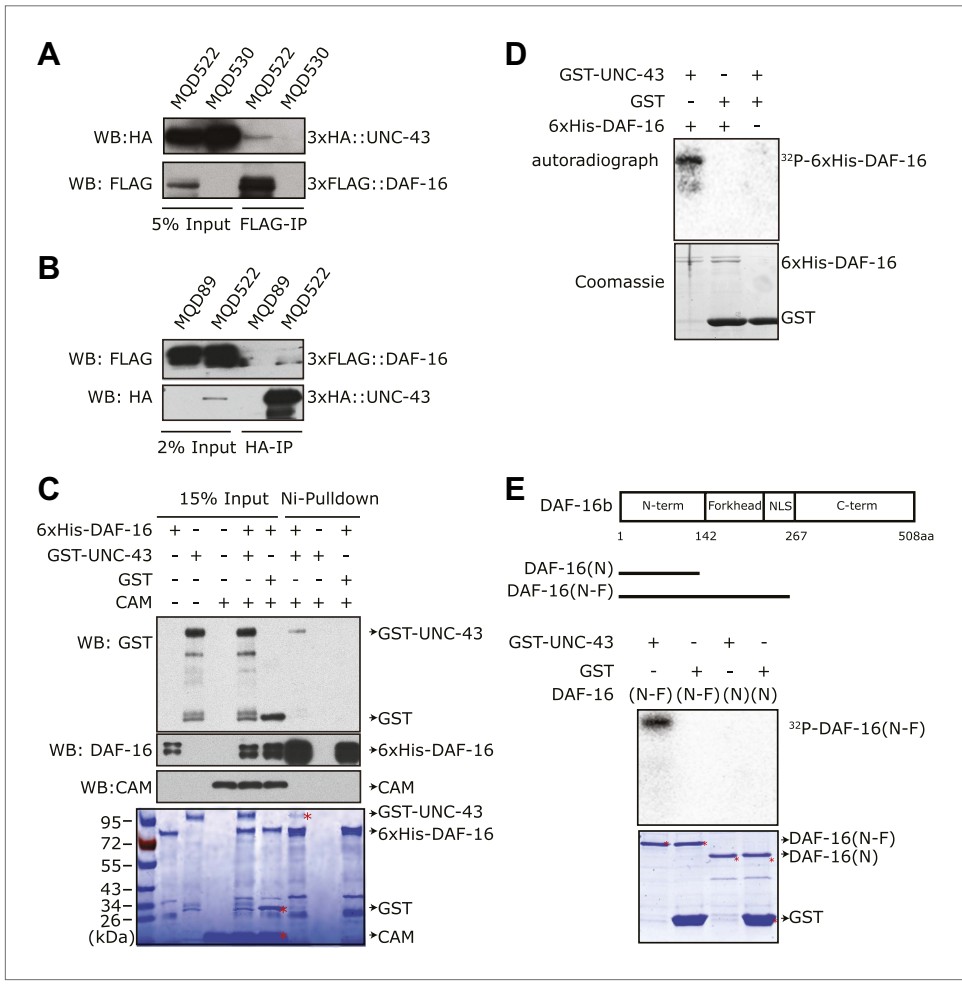

**Figure 4**. UNC-43 directly binds to and phosphorylates DAF-16. (**A**) 3xHA::UNC-43 co-immunoprecipitated with 3xFLAG::DAF-16 and (**B**) vice versa from lysates of transgenic *C. elegans* expressing both proteins. The transgenic strains are MQD522 (co-expressing 3xHA::UNC-43 and 3xFLAG::DAF-16), MQD530 (expressing 3xHA::UNC-43), and MQD89 (expressing 3xFLAG::DAF-16). (**C**) UNC-43 can directly bind to DAF-16. Purified GST-UNC-43 but not CAM or the GST control was pulled down by Ni-NTA beads through its interaction with 6xHis-DAF-16. A Coomassie gel is shown at the bottom. (**D**)–(**E**) In vitro kinase assays in the presence of [$^{32}$P]-γ-ATP, in which purified GST-UNC-43 directly phosphorylated His-tagged DAF-16 (**D**) or DAF-16 (N-F) fragments (**E**). The Coomassie-stained gel is shown below the autoradiograph.

GST-UNC-43 but not GST or calmodulin (CAM), was pulled down by nickel beads via 6xHis-DAF-16 (*Figure 4C*).

We then asked whether UNC-43 phosphorylates DAF-16. We performed an in vitro kinase assay using purified GST-UNC-43, Ca$^{2+}$/CAM, and 6xHis-DAF-16 in the presence of [$^{32}$P]-γ-ATP. GST-UNC-43 readily phosphorylated DAF-16 in vitro (*Figure 4D*). To map the phosphorylation site(s), truncated DAF-16 proteins were used as kinase substrates. UNC-43 phosphorylated the DAF-16 N-F fragment (1–267 aa) but not the N-terminal fragment (1–142 aa) (*Figure 4E*), suggesting that the phosphorylation site(s) resides in the region containing the forkhead domain (143–267 aa) and/or the C-terminal region (268–508 aa). A mass spectrometry (MS) analysis of the full-length 6xHis-DAF-16 after the in vitro kinase reaction identified two DAF-16 residues T240 and S286 that were phosphorylated by UNC-43 (*Figure 5—figure supplement 1*, *Figure 5A*). T240 and S286 are among the predicted CAMKII sites (RXXS/T or S/TXD). To confirm the MS result, we made two antibodies, one that specifically recognized phospho-T240 and one that specifically recognized phospho-S286. Using these antibodies, we verified that DAF-16 was indeed phosphorylated by UNC-43 at T240 and S286 in vitro (*Figure 5B–C*).

To confirm whether DAF-16 is phosphorylated in vivo at both sites in an UNC-43-dependent manner, we immunoprecipitated phospho-DAF-16::6xHis::GFP from *daf-16(null);unc-43(gf)* or *daf-16(null)* mutants using the phospho-DAF-16-specific antibodies and followed with anti-GFP immunoblotting. Interestingly, there was a marked increase of DAF-16 S286 phosphorylation, but not T240 phosphorylation, in the *unc-43(gf)* mutant (*Figure 5D–E*). S286A and T240A mutations completely abolished phosphorylation on the respective sites (*Figure 5C–E*). We thus concluded that UNC-43 phosphorylates DAF-16 specifically on S286. Apparently, T240 phosphorylation by UNC-43 does not occur in vivo under the tested condition.

## TAX-6 dephosphorylates DAF-16 at S286 in vitro

To test whether TAX-6 dephosphorylates DAF-16 at S286, we phosphorylated recombinant DAF-16 in vitro with UNC-43 and then incubated it with purified TAX-6•GST-CNB-1. As shown in *Figure 5F*, phosphorylation of DAF-16 at S286 was dramatically reduced by TAX-6•GST-CNB-1, while phosphorylation of T240 remained unchanged.

## Phosphorylation of DAF-16 S286 mediates the effect of UNC-43 and TAX-6

To validate the role of S286 in UNC-43- and TAX-6-regulated DAF-16 localization and longevity, we introduced the DAF-16(S286A)::6xHis::GFP transgene into *daf-16(mu86);unc-43(gf)* and *daf-16(mu86);tax-6(lf)* mutants. While DAF-16::6xHis::GFP accumulated in the nuclei of *tax-6(lf)* and *unc-43(gf)* mutants, DAF-16(S286A)::6xHis::GFP distributed diffusely throughout the cell in the same mutant backgrounds (*Figure 6A–B*), and the pattern resembled that of DAF-16::6xHis::GFP in WT worms (*Figure 1D*). In *C. elegans* whose WT copy of DAF-16 was replaced by DAF-16(S286A), the lifespan extension by either *unc-43(gf)* or *tax-6(lf)* was greatly suppressed (*Figure 6C–D*, *Figure 6—source data 1A,B*). Thus, the S286 residue is required for DAF-16-dependent effects of *unc-43(gf)* and *tax-6(lf)*, verifying that S286 is the regulatory site. In comparison, the constitutive nuclear localization phenotype of DAF-16(T240A) contradicts with the idea that UNC-43 phosphorylates T240 to promote DAF-16 nuclear accumulation (*Figure 6—figure supplement 1*). Rather, it corroborates the observation that UNC-43 does not phosphorylate T240 in vivo (*Figure 5E*).

To mimic phosphorylation by UNC-43, we mutated DAF-16 S286 to aspartic acid. When expressed in *daf-16(mu86)* animals, the DAF-16(S286D)::6xHis::GFP protein accumulated in the nucleus, whereas the WT DAF-16::6xHis::GFP protein was cytoplasmic, as expected (*Figure 6E*). Moreover, the phosphomimetic of DAF-16, but not the WT version, significantly increased the *C. elegans* lifespan in *daf-16(null)* animals (*Figure 6F*, *Figure 6—source data 1C*). The effect was not due to higher protein amounts of DAF-16(S286D) because DAF-16(S286D)::6xHis::GFP and DAF-16::6xHis::GFP were expressed at similar levels in these transgenic animals (*Figure 6E*, *Figure 6—figure supplement 2*).

Taken together, the S286D mutation mimicked the effect of *unc-43(gf)* and *tax-6(lf)* on DAF-16, whereas the S286A mutation of DAF-16 blocked the effects of *unc-43(gf)* and *tax-6(lf)*. Thus, phosphorylation of S286 is critical for CAMKII and Calcineurin to regulate DAF-16 localization and longevity.

## UNC-43 transmits signals induced by heat stress, oxidative stress, and starvation to DAF-16

S286 is different from the known AKT phosphorylation sites (T54, S238, T240, and S312 for DAF-16b) of DAF-16 (*Lin et al., 2001*). Therefore, UNC-43 and TAX-6•CNB-1 most likely respond to conditions

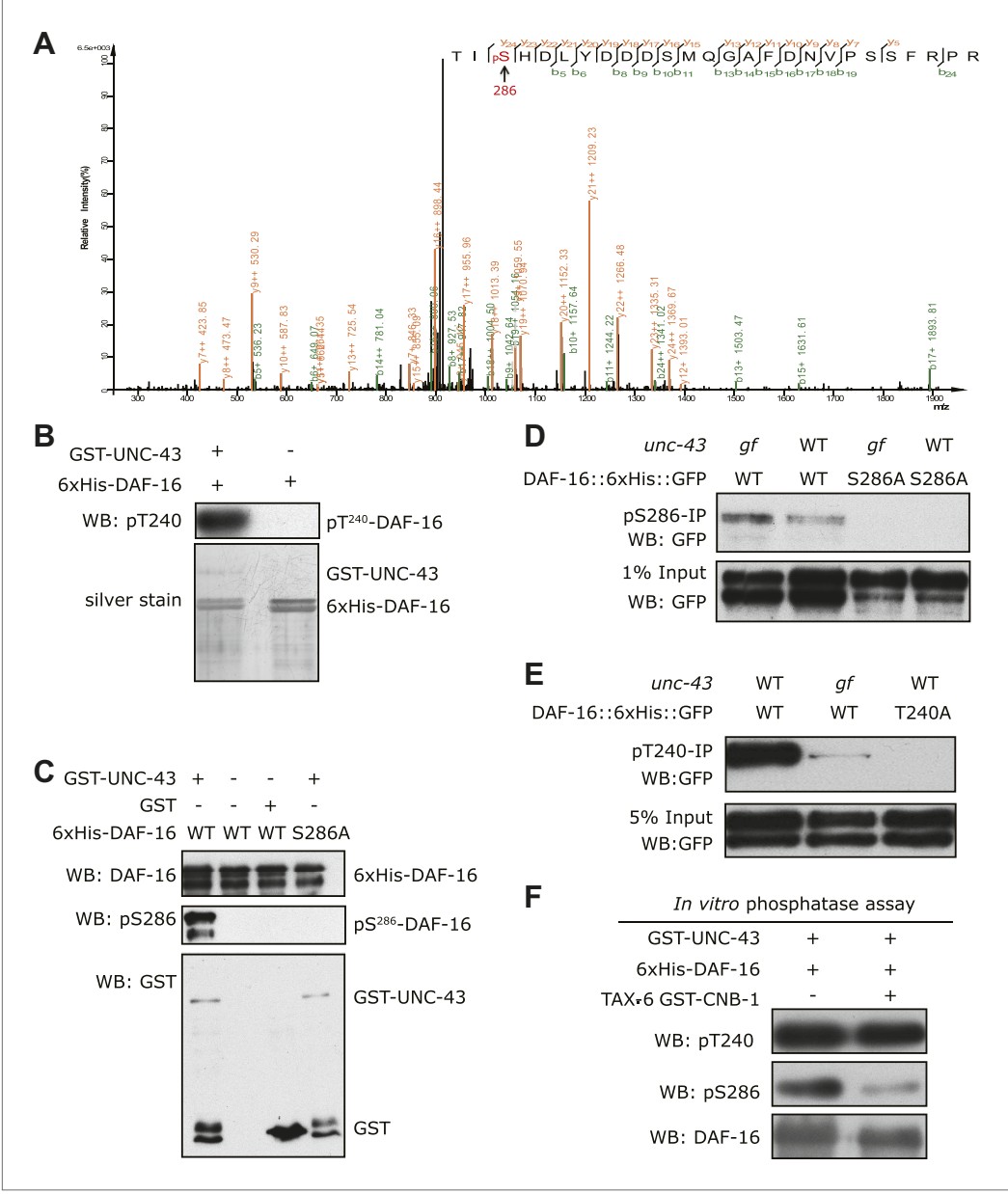

**Figure 5**. UNC-43 phosphorylates S286 of DAF-16, and TAX-6•CNB-1 removes this modification. (**A**) A mass spectrum (neutral loss-triggered MS3) of a DAF-16 peptide phosphorylated at S286 by UNC-43 in vitro. (**B** and **C**) UNC-43 in vitro kinase assays with purified WT or S286A 6xHis-DAF-16 as substrate. DAF-16 phosphorylation was detected with antibodies specific for either phospho-T240 (**B**) or phospho-S286 (**C**). (**D** and **E**) UNC-43 phosphorylates DAF-16 at S286 but not T240 in vivo. Phosphorylated DAF-16::6xHis::GFP was immunoprecipitated from *unc-43(wt)* or *unc-43(gf)* animals using antibodies specific for either phospho-S286 (**D**) or phospho-T240 (**E**) and visualized by blotting with an anti-GFP antibody. Transgenic strains expressing S286A or T240A DAF-16::6xHis::GFP served as negative controls. All strains carry the *daf-16* null allele *mu86* in the background. (**F**) TAX-6•CNB-1 dephosphorylates DAF-16 specifically at S286. Purified 6xHis-DAF-16 was phosphorylated by UNC-43 in vitro and then incubated with purified TAX-6•GST-CNB-1 after heat inactivation of UNC-43. Phospho-T240, phospho-S286, and total DAF-16 levels were assayed by western blotting.

The following figure supplements are available for figure 5:

**Figure supplement 1**. Mass spectrum showing a DAF-16 peptide phosphorylated at T240, one of the sites phosphorylated by UNC-43 in vitro.

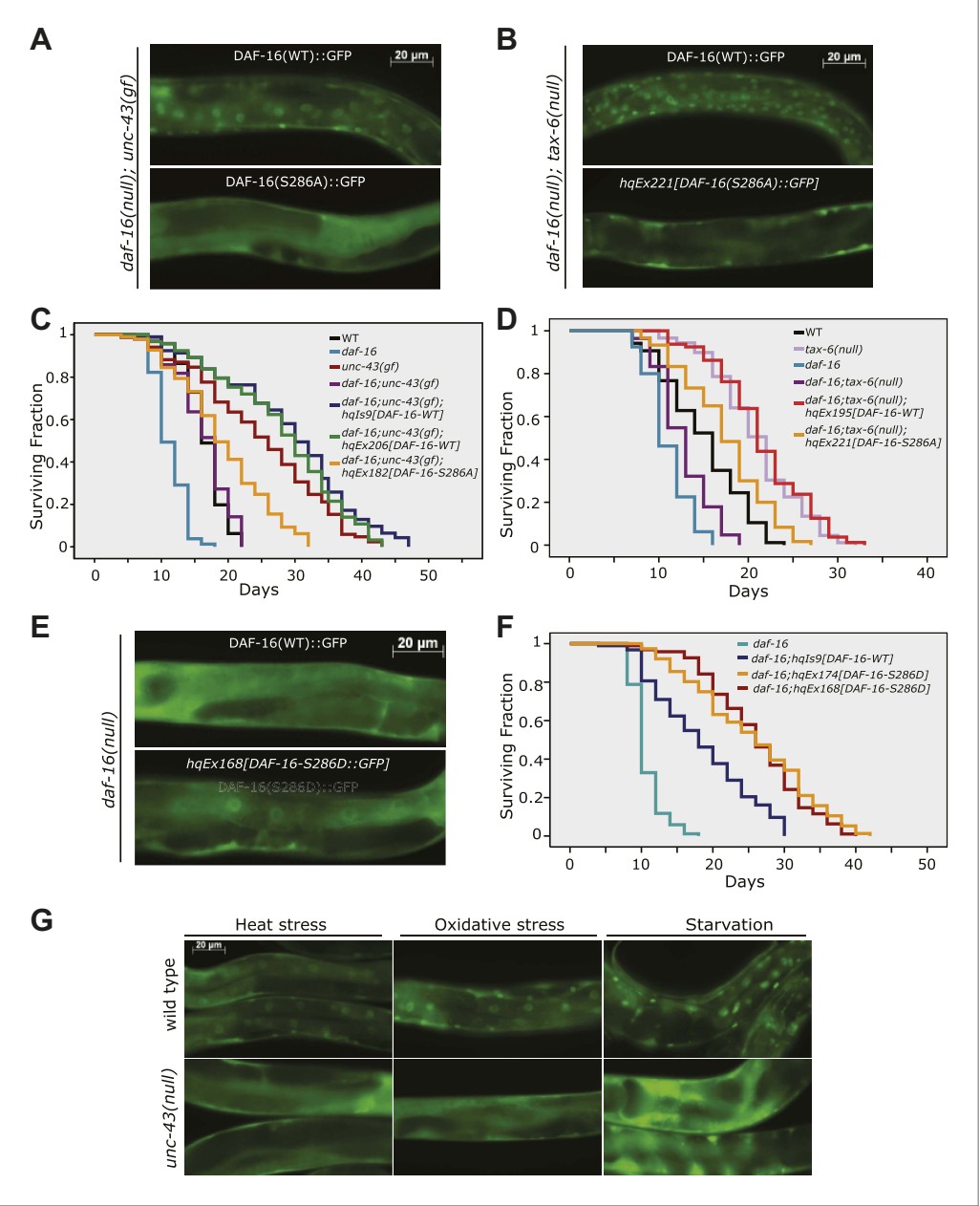

**Figure 6**. Phosphorylation of DAF-16 at S286 induces the nuclear accumulation of DAF-16 and extends lifespan. The S286A mutation prevented DAF-16::6xHis::GFP from accumulating in the nucleus in *unc-43(n498gf)* (**A**) or *tax-6(ok2065)* animals (**B**). Unlike the *DAF-16::6xHis::GFP* transgene, *DAF-16(S286A)::6xHis::GFP* failed to restore the long lifespan in *daf-16(mu86);unc-43(n498gf)* (**C**) or *daf-16(mu86);tax-6(ok2065)* mutants (**D**). p<0.001 for *unc-43(gf)* vs *daf-16;unc-43(gf);hqEx182* in (**C**) and *tax-6(null) vs. daf-16;tax-6(null);hqEx221* in (**D**). The S286D mutation caused DAF-16::6xHis::GFP to accumulate in the nucleus (**E**) and extended lifespan in the *daf-16(mu86)* background (**F**). p<0.001, *daf-16;hqEx174/168 vs. daf-16;hqIs9* in (**F**). 2 hr of heat stress at 28°C, 5 min of paraquat treatment, or 20 hr of food deprivation induced nuclear accumulation of DAF-16::6xHis::GFP in the WT animals but not the *unc-43(n498n1186)* mutants (**G**).

The following source data and figure supplements are available for figure 6:

**Source data 1**. The DAF-16(S286A) mutation largely suppressed the longevity induced by either *unc-43(gf)* or *tax-6(null)*, while the DAF-16(S286D) mutation extended lifespan.

**Figure supplement 1**. UNC-43 does not phosphorylate T240 in vivo to promote DAF-16 nuclear accumulation.

*Figure 6. Continued on next page*

*Figure 6. Continued*

**Figure supplement 2**. Similar expression levels of DAF-16(S286D)::6xHis::GFP and DAF-16::6xHis::GFP in three strains used in **Figure 6F**.

that are distinct from the conditions sensed by AKT in the IIS pathway. In addition to mutations that reduce insulin signaling, various stress conditions, including thermal stress, oxidative stress, and starvation, can trigger translocation of DAF-16 to the nucleus (*Henderson and Johnson, 2001*; *Lin et al., 2001*). Among the DAF-16 target genes are a wide range of stress response genes, including small heat shock proteins, a superoxide dismutase, catalases, and glutathione S-transferases (*Murphy et al., 2003*; *Dong et al., 2007*). Transcriptional activation of these target genes by nuclear DAF-16 helps the organism cope with stress, and *daf-16* mutant worms survive stressful conditions less well than WT worms (*Wolff et al., 2006*).

To test whether UNC-43 is required for transmitting stress signals to DAF-16, we crossed *DAF-16::6xHis::GFP* into *unc-43(null)* and compared these animals to WT animals carrying the same transgene. At 20°C, DAF-16::6xHis::GFP displayed a cytoplasmic pattern in either WT or *unc-43(null)* animals. When these worms were transferred to 28°C, DAF-16::6xHis::GFP accumulated in the nucleus in WT worms but remained cytoplasmic in *unc-43(null)* animals (*Figure 6G*). Similarly, starvation and paraquat, a chemical that generates reactive oxygen species (ROS) inside the cells, caused DAF-16 to translocate to the nucleus in WT but not *unc-43(null)* worms (*Figure 6G*). The results demonstrated that *unc-43* is required for nuclear accumulation of DAF-16 in response to different types of stress.

## Crosstalk between insulin signaling and parallel CAMKII/Calcineurin signaling

The site of DAF-16 that is phosphorylated by CAMKII is different from the site that is phosphorylated by AKT. Moreover, phosphorylation by AKT and CAMKII has opposite effects on DAF-16 activity (*Figures 5 and 6* and *Lin et al., 2001*). Thus, the IIS pathway (DAF-2) and the CAMKII pathway might work in parallel. In agreement with this hypothesis, reduced IIS and increased phosphorylation of DAF-16 S286, either by *unc-43(gf)* or *tax-6(lf)*, displayed an additive effect on lifespan (*Figure 7A–B*, *Figure 7—source data 1A*). These two pathways, however, also seem to crosstalk. First, an *unc-43* null allele shortened the *daf-2* lifespan by 31% (*Figure 7C*, *Figure 7—source data 1B*), indicating that *unc-43* is required for full lifespan extension by *daf-2(e1370)*. This result suggests that part of the signaling from DAF-2 is mediated by UNC-43. However, phosphorylation of DAF-16 S286 seems nonessential for the longevity of *daf-2* animals (*Figure 7—figure supplement 1*, *Figure 7—source data 1C*), suggesting that an additional UNC-43 target(s) is involved in DAF-2 signaling. Second, in the constitutively active *unc-43(gf)* mutant, which has increased phosphorylation on DAF-16 S286 (*Figure 5D*), phosphorylation by AKT on T240 is greatly reduced (*Figure 5E*). Therefore, in long-lived *unc-43(gf)* animals, phosphorylation on these two residues appears to be coordinated, but it is unclear how this coordination occurs. The AKT kinase activity may be reduced in *unc-43(gf)*, or phospho-S286 DAF-16 may be a poor substrate for AKT. In either case, dampened phosphorylation on the AKT site may be expected to prevent DAF-16 from forming a complex with the 14-3-3 protein, which has been shown to bind to and sequester AKT-phosphorylated FOXO (DAF-16 or mammalian FOXO3) in the cytoplasm (*Brunet et al., 1999*; *Berdichevsky et al., 2006*; *Li et al., 2007*). Consistently, we found that *unc-43(gf)* reduced the amount of DAF-16-associated 14-3-3 without affecting the total amount of 14-3-3 (*Figure 7—figure supplement 2*). It remains to be seen whether S286 phosphorylation directly or indirectly promotes DAF-16 nuclear accumulation.

While IIS and CAMKII/Calcineurin signaling act largely in parallel with each other, it is unclear what would happen if one pathway was in a state to inhibit DAF-16 while the other pathway was in a state that would activate it. To genetically mimic this scenario, we combined *daf-2* RNAi with either *unc-43(null)* or *tax-6(jh107)*, a constitutively active *tax-6(gf)* allele in which the autoinhibitory domain was deleted (*Lee et al., 2004*). DAF-16::GFP was cytoplasmic in *unc-43(null)* or *tax-6(gf)*, but nuclear DAF-16::GFP was clearly visible above the cytoplasmic background in *daf-2(RNAi)*, *daf-2(RNAi);unc-43(null)*, or *daf-2(RNAi);tax-6(gf)* mutants (*Figure 8A*). We also combined *daf-18* RNAi, which enhances IIS to inhibit DAF-16 (*Ogg and Ruvkun, 1998*), with *unc-43(gf)* or *tax-6(lf)*, which activates DAF-16. In both

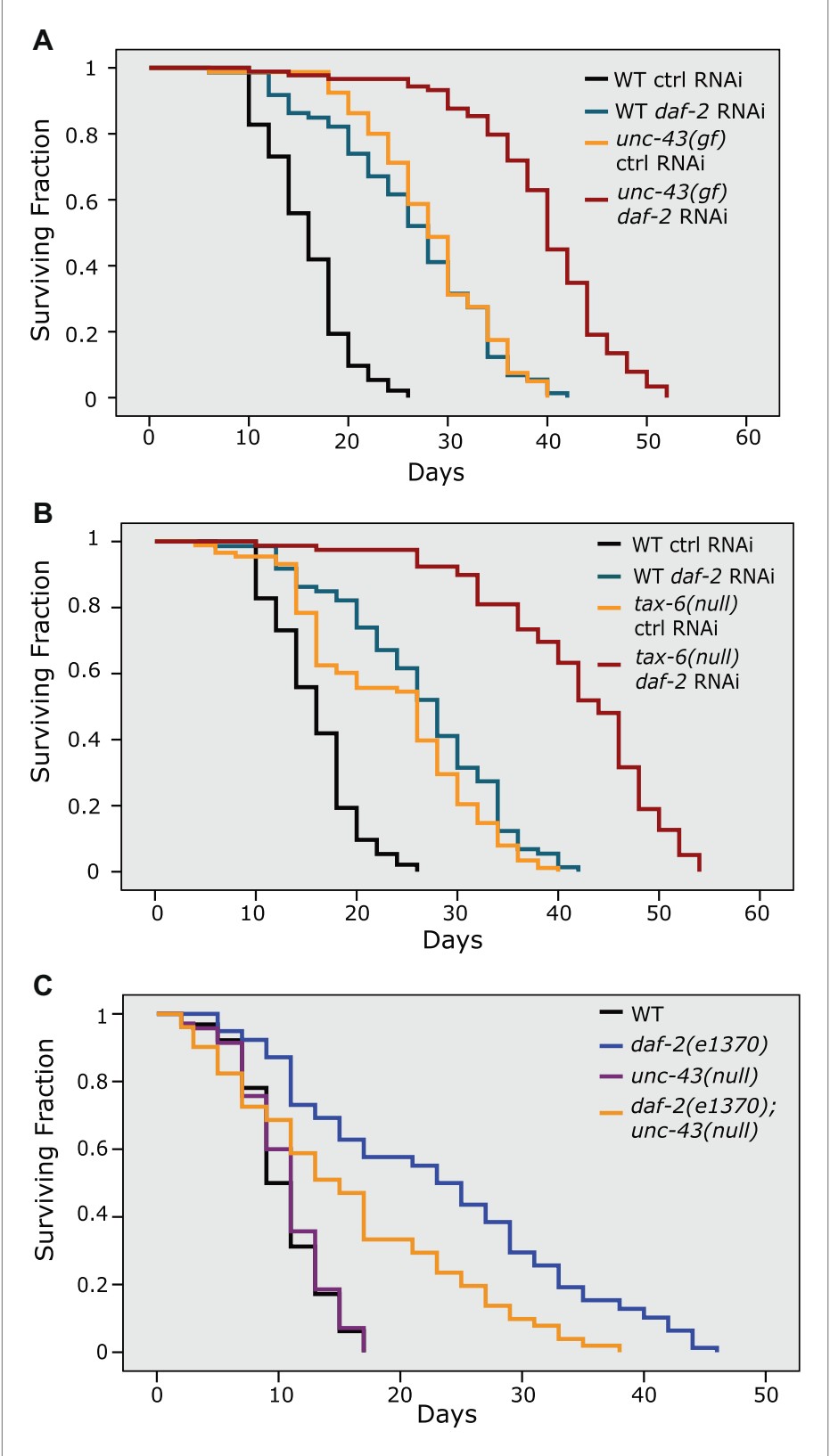

**Figure 7**. The longevity of *daf-2* mutants was partially suppressed by *unc-43(null)* and further enhanced by *unc-43(gf)* or *tax-6(null)*. *unc-43(n498)*, a gain-of-function allele (**A**), and *tax-6(ok2065)*, a null allele (**B**), each extended lifespan in the WT background and further enhanced the longevity of *daf-2(RNAi)* animals. (**C**) The

*Figure 7. Continued on next page*

*Figure 7. Continued*

*daf-2(e1370);unc-43(n498n1186)* double mutant lived a significantly shorter life than *daf-2(e1370)*, while the *unc-43(n498n1186)* mutant had a WT-like lifespan. In (**A**) and (**B**), p<0.001 for WT vs *tax-6(null)* or *unc-43(gf)*, *unc-43(gf);daf-2(RNAi)* vs. *daf-2(RNAi)* or *unc-43(gf)*, and *tax-6(null);daf-2(RNAi)* vs. *daf-2(RNAi)* or *tax-6(null)*. In (**C**), p<0.001 for *daf-2* vs. *daf-2;unc-43(null)*, p=0.61 for *unc-43(null) vs.* WT in a lifespan assay at 25°C.

The following source data and figure supplements are available for figure 7:

**Source data 1**. The long lifespan of *daf-2(RNAi)* animals was further extended by either *unc-43(gf)* or *tax-6(null)*, but shortened by *unc-43(null)*.

**Figure supplement 1**. The DAF-16 S286A mutation appears not to affect daf-2 longevity.

**Figure supplement 2**. In the unc-43(n498gf) mutant, in which DAF-16 accumulates in the nucleus, the amount of DAF-16/14-3-3 complex is reduced.

cases, the nuclear accumulation induced by *unc-43(gf)* or *tax-6(lf)* was greatly diminished by *daf-18* RNAi (**Figure 8B**). Therefore, under conditions in which there is a conflict between IIS and CAMKII/Calcineurin signaling toward DAF-16, the insulin signaling pathway appears to dominate.

Based on the experimental evidence above, we propose a model (**Figure 8C**) in which DAF-16 integrates hormonal signals and stress signals transmitted by the IIS pathway (from DAF-2 to AKT) and CAMKII (UNC-43), respectively. CAMKII and the counterbalancing phosphatase Calcineurin (TAX-6•CNB-1) act in parallel and coordinately with the IIS pathway to regulate DAF-16 activity.

## Mammalian CAMKII and Calcineurin also regulate phosphorylation of FOXO3 at a conserved serine residue

Sequence analysis of FOXO proteins revealed that DAF-16 S286, the CAMKII phosphorylation site, is widely conserved in nematode species, zebra fish, frogs, mice, and humans (**Figure 9A**). It is equivalent to S298 of mouse FOXO3 (mFOXO3) and S299 of human FOXO3 (hFOXO3) (**Figure 9A**), and may be the equivalent of S303 of human FOXO1 or S300 of mouse FOXO1 (not shown). Such conservation suggests that mammalian FOXO proteins are likely regulated by CAMKII and Calcineurin.

To determine if this is true, we first expressed a FLAG-tagged mouse FOXO3 together with a GFP-tagged mouse CAMKII alpha (CAMKIIA) or beta (CAMKIIB) isoform, or a Myc-tagged human Calcineurin A (CnA) in HEK293T cells, and found that CAMKII and CnA both interact with FOXO3 (**Figure 9B–C**). Next, we asked whether CAMKII phosphorylates FOXO3 in vitro. Purified GST fusion proteins containing five non-overlapping fragments of FOXO3 (**Lehtinen et al., 2006**) were incubated with an active recombinant human CAMKIIA. CAMKIIA robustly phosphorylated the P2 and P3 fragments but not the others (**Figure 9D**), placing the phosphorylation sites within aa 154–409. Mass spectrometry and mutagenesis studies of the full-length mFOXO3 or the P2 and P3 fragments from the in vitro kinase assay mapped the CAMKIIA phosphorylation sites to S252, S279, and S298 (not shown). The same sites were also identified from FLAG-mFOXO3 co-expressed with CAMKIIA in HEK293T cells (**Figure 9E**, **Figure 9—figure supplement 1**). However, only mFOXO3 S298, which corresponds to DAF-16 S286, had a significant increase in phosphorylation when cells were co-transfected with CAMKIIA (**Figure 9F**). This shows that S298 is the principle CAMKIIA phosphorylation site in vivo. S252 in mouse FOXO3 or S253 in human FOXO3, the counterpart of DAF-16 T240 (**Figure 9A**), is a highly conserved AKT phosphorylation site (**Brunet et al., 1999**). Using a specific antibody, we found that neither CAMKIIA nor Calcineurin regulated phosphorylation of endogenous human FOXO3 at S253 in HEK293T cells (**Figure 9—figure supplement 2**). Moreover, Calcineurin can reduce the phosphorylation of the FOXO3-P3 fragment (containing S298) by CAMKIIA, but not that of the P2 fragment (containing S252) (**Figure 9—figure supplement 3**). Taken together, these results demonstrate that mammalian CAMKII and Calcineurin regulate phosphorylation of FOXO3 at the same conserved site as in *C. elegans* (S298 in mouse FOXO3 and S286 in DAF-16).

A recent study by Ozcan et al. shows that four non-AKT phosphorylation sites are involved in the activation of murine FOXO1 by CAMKIIγ, likely through the p38 MAP kinase, in glucagon-stimulated hepatocytes (**Ozcan et al., 2012**). Serine 300 of mouse FOXO1, the equivalent of S298 of mouse FOXO3, is not among them (S246, S295, S467, and S475). To find out whether the CAMKIIA isoform

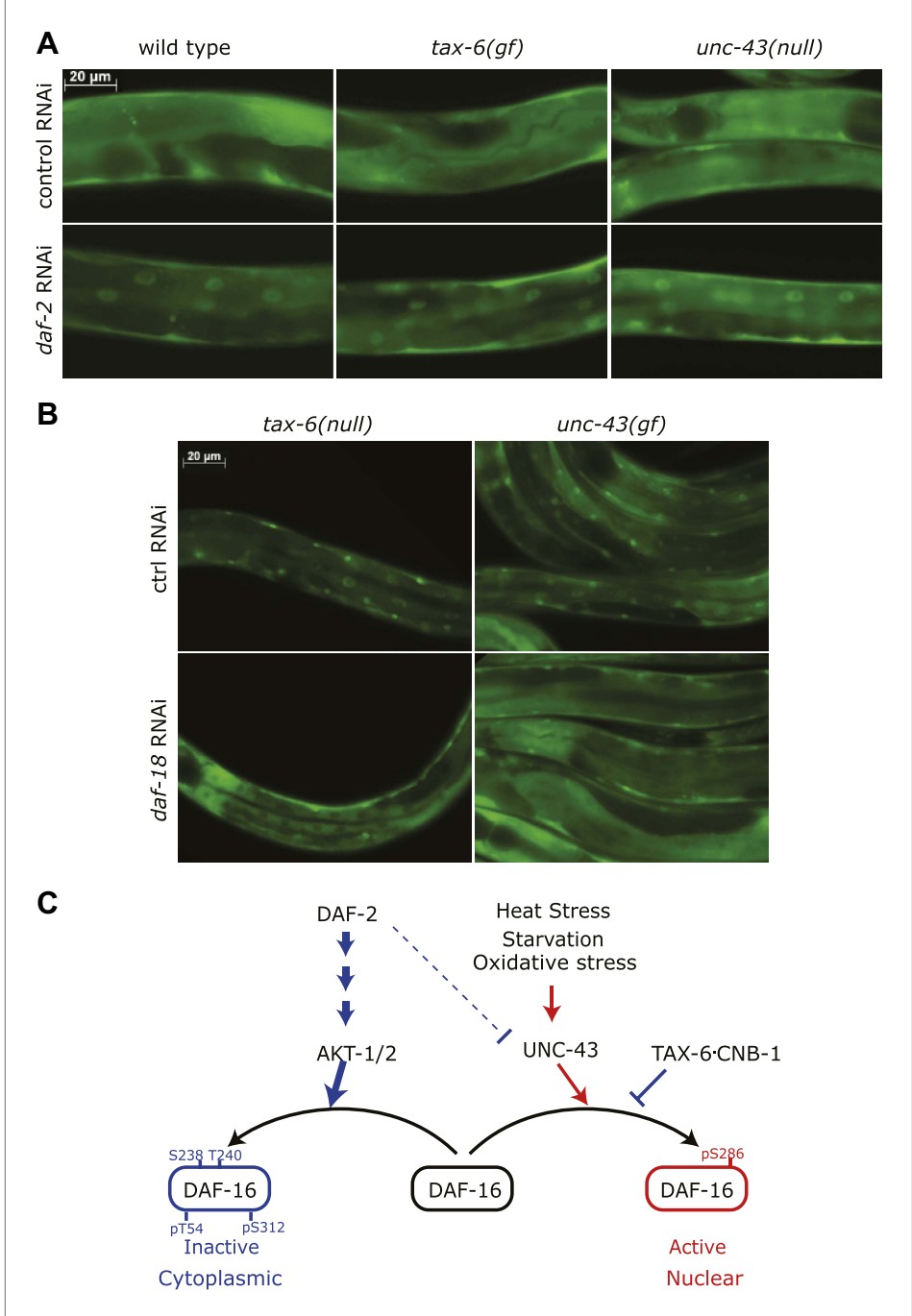

**Figure 8**. Insulin signaling overpowers CAMKII and Calcineurin in regulation of DAF-16 localization in *C. elegans*. (**A**) *tax-6(gf)* or *unc-43(null)* failed to abolish DAF-16 nuclear accumulation induced by *daf-2(RNAi)*. (**B**) *tax-6(null)* or *unc-43(gf)* failed to overcome the inhibition of DAF-16 nuclear localization by *daf-18(RNAi)*. (**C**) A model showing the regulation of DAF-16 (FOXO) by insulin signaling, UNC-43 (CAMKII), and TAX-6•CNB-1 (Calcineurin).

phosphorylates FOXO1 in the same way as it does FOXO3, we expressed a FLAG-tagged human FOXO1 in HEK293T cells co-transfected or not with CAMKIIA. Mass spectrometry analysis identified three CAMKIIA-regulated phosphorylation sites equivalent to mouse FOXO1 S295, S467, and S475 (not shown). This agrees with the findings by Ozcan et al. and suggests that CAMKII directly phosphorylates FOXO3 but not FOXO1 in mammals.

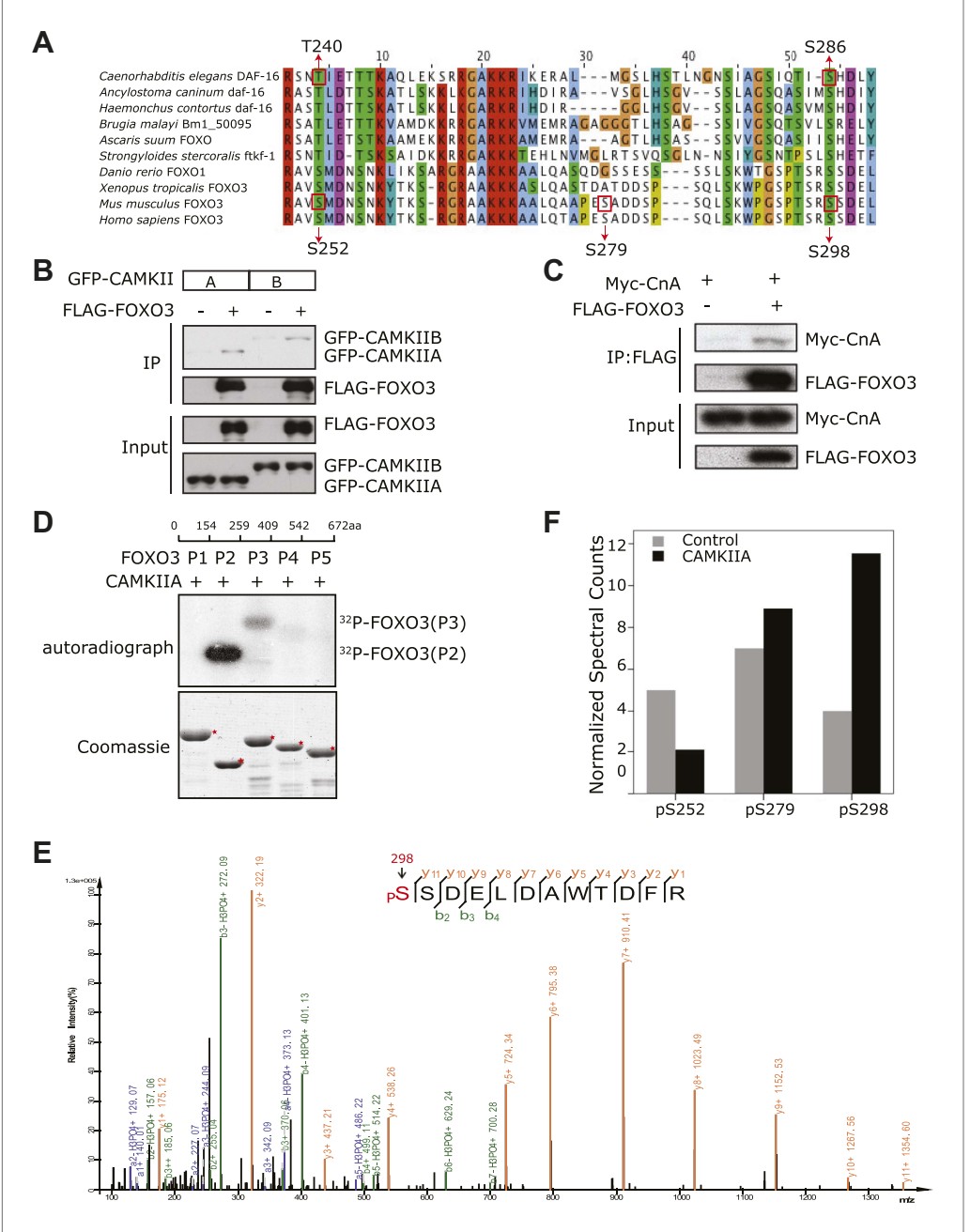

**Figure 9**. CAMKII and Calcineurin regulate phosphorylation of mouse FOXO3 at S298. (**A**) Sequence alignment of selected FOXO homologs from nematode species to human. The region containing DAF-16 T240 and S286 is shown. T240 is conserved in all FOXO homologs. S286 is less conserved, but it is found in many nematode FOXOs and a number of vertebrate FOXOs. The DAF-16 or mouse FOXO3 residues phosphorylated in vitro by UNC-43 or CAMKIIA are boxed, with amino acid positions shown above or below. GFP-tagged mouse CAMKIIA or CAMKIIB, or Myc-tagged human Calcineurin A was transfected either alone or with FLAG-tagged mouse FOXO3 into HEK293T cells. Immunoprecipitation of FLAG-FOXO3 pulled down both CAMKII isoforms (**B**) and Calcineurin (**C**). (**D**) Two FOXO3 fragments P2 and P3, corresponding to amino acids 154–259 and 259–409, were phosphorylated in vitro by CAMKIIA in the presence of [$^{32}$P]-γ-ATP. The input protein substrates are shown at the bottom. (**E**) Mass spectrum showing a FOXO3 peptide phosphorylated by CAMKIIA at S298. (**F**) CAMKIIA phosphorylates FOXO3 at S298 in vivo. FLAG-FOXO3 immunoprecipitated from HEK293T cells co-transfected or not with CAMKIIA was digested with trypsin and subjected to mass spectrometry analysis. Spectral counts (i.e., number of observations) of the indicated phospho-S/T are normalized to the total spectral counts of FLAG-FOXO3.

*Figure 9. Continued on next page*

*Figure 9. Continued*

The following figure supplements are available for figure 9:

**Figure supplement 1**. Analysis of FOXO3 phosphorylation sites in vivo.

**Figure supplement 2**. CAMIIA had no effect on S253 phosphorylation of human FOXO3 (corresponding to S252 of mouse FOXO3) in vivo.

**Figure supplement 3**. In vitro Calcineurin phosphatase assay on FOXO3.

Lastly, we determined whether CAMKII and Calcineurin regulate the transcriptional activity of FOXO3. Mammalian CAMKII can be activated by autophosphorylation at threonine 286 (*Mukherji et al., 1994*; *Rich and Schulman, 1998*). Mutation of this residue to A or D results in a kinase dead or constitutively active form of CAMKII, respectively (*Fong et al., 1989*). Using a luciferase reporter assay, we found that the transcriptional activity of FOXO3 was indeed stimulated by CAMKIIA in a way that required S298 of FOXO3 (*Figure 10A*) and T286 of CAMKIIA (*Figure 10B*), and further enhanced by the constitutively active CAMKIIA mutant (*Figure 10C*). Consistently, expression of Calcineurin significantly inhibited the transcriptional activity of FOXO3 (*Figure 10D*). From these results, we conclude that CAMKIIA and Calcineurin oppose each other in regulating the transcriptional activity of mouse FOXO3 through phosphorylation or dephosphorylation at S298. This mechanism is conserved from *C. elegans* to mammals.

## Discussion

### CAMKII and Calcineurin constitute a previously uncharacterized signaling branch that targets DAF-16

Our data suggest that CAMKII (UNC-43) and Calcineurin (TAX-6•CNB-1), by phosphorylating or dephosphorylating serine 286 of DAF-16, extend or shorten the lifespan of *C. elegans*. The direct, antagonistic nature of these two enzymes also raised an interesting possibility that, with their opposing effects on DAF-16 and a common activating signal ($Ca^{2+}$/CAM), UNC-43 and TAX-6•CNB-1 would constitutively negate each other and have essentially no control of DAF-16. However, this outcome most likely does not occur because of the difference in the spatial distribution and temporal activation profiles of the two enzymes. TAX-6 and CNB-1, similar to DAF-16, are expressed in most cells in *C. elegans* (*Ogg et al., 1997*; *Bandyopadhyay et al., 2002*; *Kuhara et al., 2002*; *Dong et al., 2007*). In contrast, *unc-43* promoter-driven GFP was only detected in neurons and the intestine (*Hunt-Newbury et al., 2007*). We also detected UNC-43 primarily in the nervous system (not shown).

At the subcellular level, the TAX-6::GFP fusion protein is distributed diffusely throughout the cytoplasm and inside the nucleus (*Kuhara et al., 2002*; *Dong et al., 2007*). In contrast, the CFP::UNC-43 fusion protein is concentrated on perinuclear structures and clusters in neurites (*Umemura et al., 2005*). Temporally, $Ca^{2+}$-activated CAMKII phosphorylates another molecule of CAMKII at T286 to convert it to a $Ca^{2+}$/CAM-independent kinase, prolonging the kinase activity after the initial activation (*Erickson et al., 2011*). CAMKII can also be activated by ROS at low $Ca^{2+}$ concentrations (*Erickson et al., 2011*). In mammalian cells, inhibition of Calcineurin by cytoplasmic CAMKII and inhibition of CAMKII by Calcineurin have both been reported (*MacDonnell et al., 2009*; *Kubokawa et al., 2011*), suggesting that the balance is intricately maintained. Therefore, in the case of the simultaneous activation of CAMKII and Calcineurin by $Ca^{2+}$/CAM, either the kinase or the phosphatase could dominate the regulation of DAF-16 either locally (e.g., cytoplasm vs nucleus) or for certain time windows. This balance likely fine-tunes the DAF-16 activity in response to a changing environment.

### CAMKII responds to a variety of stress signals

It is likely that UNC-43/Calcineurin is responsible for activating DAF-16 when worms experience a variety of stress signals, including heat stress, oxidative stress, or prolonged starvation (*Figure 6G*). Food deprivation acts through UNC-43 to suppress spontaneous sex-muscle contraction in *C. elegans* males (*LeBoeuf et al., 2007*). Reported in HEK293T cells, human FOXO3 is phosphorylated at non-AKT sites under stress conditions such as heat shock or $H_2O_2$ treatment (*Brunet et al., 2004*). ROS-induced oxidation of two methionine residues activates CAMKII, and this activation can occur

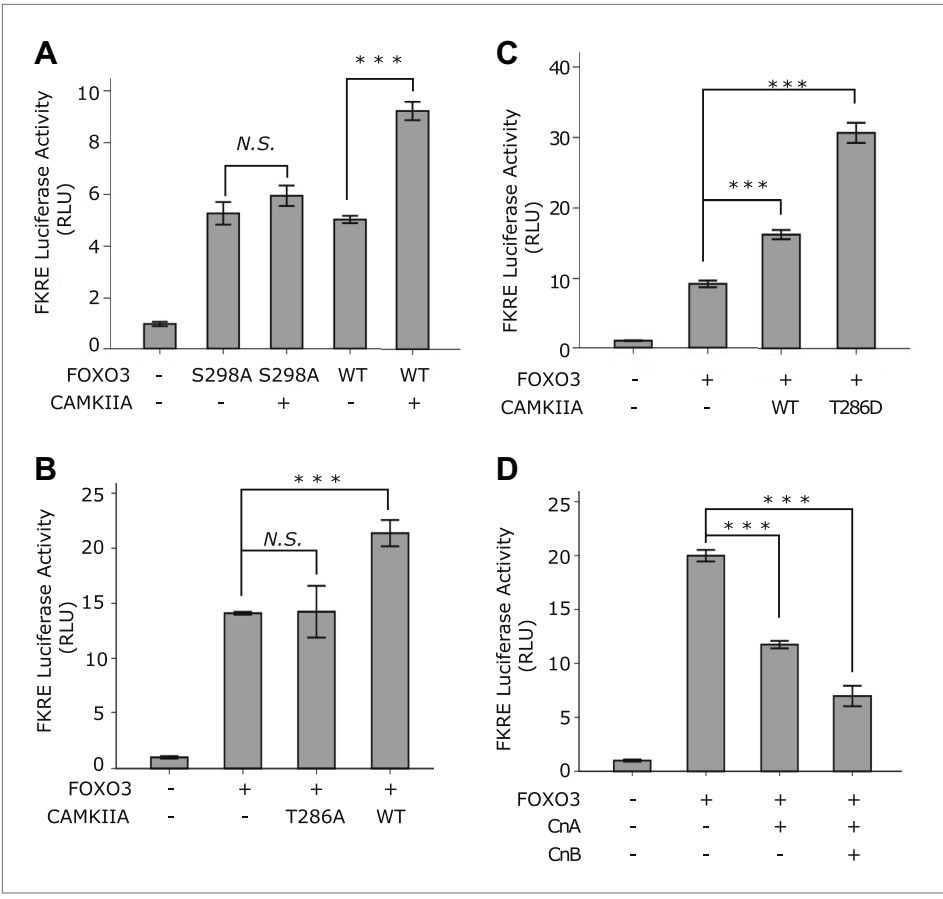

**Figure 10**. CAMKII and Calcineurin regulate the transcriptional activity of FOXO3. (**A**)–(**D**) HEK293T cells were transfected with the indicated constructs together with a 3xIRS-firefly luciferase reporter and a TK-renilla luciferase reporter. Mean ± SD of firefly/renilla luciferase activity relative to the empty vector transfection is plotted (***p<0.0001, Student's *t*-test, n = 3; *N.S.* for not significant). (**A**) Co-transfection of CAMKIIA stimulated the transcriptional activity of WT FOXO3 but not the FOXO3 (S298A) mutant. (**B**) WT but not an inactive T286A mutant CAMKIIA, transcriptionally activated FOXO3. (**C**) A constitutively active T286D mutant CAMKIIA further enhanced FOXO3 activity compared to WT CAMKIIA. (**D**) FOXO3 was transcriptionally inhibited by Calcineurin A, and further inhibited if Calcineurin A and Calcineurin B were both expressed.

under conditions of low $Ca^{2+}$ (*Erickson et al., 2011*). Thus, CAMKII is perfectly equipped to transmit ROS signaling and calcium signaling, and phosphorylation of FOXO3 by CAMKII may play a highly conserved role in the stress response, from worms to humans.

## Daf-16 is not the only substrate of worm Calcineurin that regulates lifespan

To date, DAF-16 is the only biochemically and genetically proven substrate of TAX-6•CNB-1, but it is not the only substrate involved in lifespan regulation by Calcineurin. Inactivation of CRTC-1, a coactivator of CREB, contributes to the longevity of *tax-6(lf)* animals (*Mair et al., 2011*). Hence, CRTC-1 is an excellent candidate, although it has not been shown that TAX-6•CNB-1 directly dephosphorylates CRTC-1. There may be additional substrates. Regardless, the discoveries made thus far jointly provide a good explanation for the large but incomplete suppression of *tax-6(lf)* longevity by *daf-16(null)* (*Dong et al., 2007*).

## Dysregulation of FOXO3 by CAMKII and Calcineurin may be linked to diabetes

Post-transplant new-onset diabetes mellitus (NODM) is a frequent and serious complication for organ transplant patients. One of the NODM risk factors is the use of Calcineurin inhibitors, such as cyclosporin A and FK506, which suppress immune-rejection of donor organs (*Heisel et al., 2004*). In light of this

work, we propose that sustained activation of FOXO3 resulting from chronic inhibition of Calcineurin may cause NODM in transplant patients. Given the conservation and broad tissue distribution of CAMKII, Calcineurin, and FOXO3, the regulatory mechanism uncovered in this study likely plays a role in many physiological functions.

# Materials and methods

## *C. elegans* strains

All strains were cultured at 20°C following standard protocols unless otherwise indicated (*Brenner, 1974*). Double mutants were made using standard genetic methods, and the genotypes were confirmed using PCR or PCR followed by sequencing. The strains and oligos used in the study are listed in *Supplementary file 1A,B*, respectively.

## Antibodies

The following antibodies were purchased: Mouse anti-Actin (Sigma-Aldrich, St Louis, MO), Mouse anti-tubulin (Sigma-Aldrich), mouse anti-CAM (Millipore, Billerica, MA), mouse anti-FLAG (Sigma-Aldrich), anti-FLAG M2 Affinity Gel (Sigma-Aldrich), rabbit anti-GFP (Abmart, Shanghai, China), mouse anti-GFP (Roche, Basel, Switzerland), rabbit anti-GFP (Invitrogen, Carlsbad, CA), rabbit anti-GST (GeneSci, Beijing, China), monoclonal anti-HA agarose (Sigma-Aldrich), rabbit anti-mFOXO3(phospho-S252)/hFOXO3(phospho-S253) (Cell Signaling, Danvers, MA), Mouse anti-Myc (Santa Cruz Biotechnology, Dallas, TX), goat anti-mouse IgG HRP (Jackson Immuno Research, West Grove, PA), and goat anti-rabbit IgG HRP (BaiHuiZhongYuan, Beijing, China). Rabbit α-HA and monoclonal mouse α-DAF-16 antibodies were made at the antibody center of the NIBS, Beijing. A single-chain anti-GFP antibody was produced as described before (*Kubala et al., 2010*) and cross-linked to NHS-activated Sepharose 4 Fast Flow beads (GE Healthcare, Piscataway Township, NJ). We called this GBP beads. To produce the DAF-16 antibody, full-length DAF-16b was cloned into the His-tag vector pET-28a, expressed in *Escherichia coli* BL21, and purified by Ni-NTA resin (Qiagen). Rabbit α-DAF-16b(pT240) and α-DAF-16b(pS286) antibodies were custom made and affinity purified at Abmart (Shanghai, China) using the synthetic DAF-16 phospho-peptides 'RERSN(pT)IETTT-C' and 'SIQTI(pS)HDLYD-C,' respectively. A Rabbit polyclonal anti-human 14-3-3 antibody that recognizes both *C. elegans* 14-3-3 proteins was a gift from Dr Yamei Wang (Xiamen University, China).

## Plasmids

The *daf-16* constructs were subcloned and further modified from a P*daf-16*::*daf-16b*::*gfp* construct described previously (the *daf-16b* isoform used to be called *daf-16a2*) (*Henderson and Johnson, 2001*). The GST-CNB-1 and TAX-6 bacterial expression constructs were modified from the GST-CNB-1 and GST-TAX-6 plasmids reported before (*Bandyopadhyay et al., 2002*). The *unc-43* expression constructs were made from *unc-43* cDNAs that were amplified from a *C. elegans* cDNA library. The constructs expressing full-length or truncated mouse FOXO3 have been described (*Lehtinen et al., 2006*), and the FOXO3-P2-S252A mutant was made by site-directed mutagenesis. The CAMKIIA and CAMKIIB expression constructs have described (*Gaudillière et al., 2004*). The Myc-CnA and Myc-CnB expression constructs under the CMV promoter were subcloned from 'pET15b CnA CnB' (Addgene). The FKRE-luciferase expression vector was kindly provided by Dr Azad Bonni (Harvard Medical School).

## Purified proteins

GST-CNB-1 and TAX-6 were co-expressed in *E. coli* BL21 and affinity purified using glutathione Sepharose (GE Healthcare) in PBS containing 0.5% NP-40 following standard protocols. GST-DAF-16(N) and GST-DAF-16(N-F) were purified in the same manner. Recombinant GST-UNC-43 was purified as described above but using a buffer containing 50 mM Tris-HCl, pH 7.6, 150 mM NaCl, 0.5% NP-40. Recombinant 6xHis-DAF-16 and 6xHis-DAF-16-6AM were expressed in BL21 and affinity purified using Ni-NTA agarose resin (Qiagen) in buffer containing 20 mM HEPES, pH 7.5, 150 mM NaCl, and 0.5% NP-40. His-tagged proteins were eluted off beads by adding 250 or 500 mM imidazole to the buffer after a 30 mM imidazole wash. Eluted proteins underwent buffer exchange and were concentrated using Centricon purifying units (Millipore) with 10-kDa (for TAX•GST-CNB-1) or 50-kDa (for other proteins) cutoffs to 0.5–1.0 mg/ml protein in storage buffer (20 mM HEPES, pH 7.5, 150 mM NaCl, 0.5% NP-40, 10% glycerol); the resulting solutions contained >1 mM glutathione or >5 mM imidazole. Bovine Calmodulin and human Calcineurin were purchased from Merck. Active human

GST-CAMKIIA was ordered from Sigma. Recombinant mFOXO3 P1-P5 fragments were purified as described (*Lehtinen et al., 2006*).

## Lifespan assay

Lifespan assays were carried out at 20°C unless otherwise indicated. To synchronize worms, twenty adult worms were allowed to lay eggs on NGM plates for 4 hr before being removed. After the progeny grew to the L4 stage, they were transferred to new plates (10 worms/plate). At least ten plates (100 worms) were used for the lifespan assay of each strain or RNAi treatment. Worms were transferred to fresh plates every 2 days until they ceased laying egg, after which worms were transferred to fresh plates every week. For lifespan assays that involved or would be compared with the *unc-43(gf)* mutant, 50 µg/ml FUDR was added to the plates to prevent its Egl (egg-laying defective) phenotype from interfering with lifespan measurement. In these experiments (*Figure 2—source data 1B*; *Figure 2—source data 1C*; *Figure 3—source data 1*; *Figure 6—source data 1A*; *Figure 6—source data 1C*; *Figure 7—source data 1A*), worms were transferred to fresh plates every 4 days until death. Live worms were scored every 2 days. Worms were considered dead if they failed to respond to gentle touches with a worm pick on the head and tail. Worms that had internally hatched larvae ('bagged') or ruptured vulvae ('exploded') or crawled off the agar surface were censored. SPSS (Statistical Package for the Social Sciences) software was used for the statistical analysis of the lifespan data, and the log-rank method was used to calculate p values.

## RNAi

RNAi assays were performed at 20°C using the feeding method as previously described (*Kamath et al., 2003*). Worms were fed RNAi bacteria from the time of hatching unless otherwise indicated. *E. coli* HT115(DE3) transformed with pAD12, an empty RNAi vector (*Arantes-Oliveira et al., 2002*), was used as a control. pTL13 is an RNAi construct targeting *unc-43*, which we generated by inserting a 332-bp cDNA fragment of *unc-43* into the multiple cloning sites of pAD12. The other RNAi bacterial strains were obtained from the Ahringer RNAi library.

## Pull down assay

To examine the interaction between DAF-16 and TAX-6, 6xHis-DAF-16 was incubated with purified TAX-6•GST-CNB-1 complex in the presence of 2 µg CAM and 30 µl pre-washed Ni-NTA agarose resin (Qiagen) in binding buffer A (50 mM Tris, pH 8.0, 100 mM NaCl, 4 mM $CaCl_2$, 2 mM $MgCl_2$, 0.5% NP-40) at 4°C for 2 hr. The beads were then washed with washing buffer A (50 mM Tris, pH 8.0, 100 mM NaCl, 4 mM $CaCl_2$, 2 mM $MgCl_2$, 0.5% NP-40, 30 mM imidazole) and eluted with SDS loading buffer. Samples were resolved by SDS-PAGE and stained with Coomassie blue.

For DAF-16 and UNC-43 binding, 6xHis-DAF-16 was incubated with purified GST-UNC-43 and CAM in the presence of 30 µl pre-washed Ni-NTA agarose in binding buffer A without $MgCl_2$. After binding, beads were washed with washing buffer A without $MgCl_2$ and eluted with SDS loading buffer. The samples were resolved by replicate SDS-PAGE gels, one for Coomassie blue or silver staining, and the other for western blots with anti-GST, anti-DAF-16 or anti-CAM antibodies.

To map the TAX-6 binding site on DAF-16, 5 µg of GST or GST-tagged DAF-16 fragments were incubated with TAX-6::GFP-containing worm lysate (~15 µg/µl proteins in 300 µl of 20 mM Tris, pH 8.0, 1% NP-40, 10% glycerol, 1x Protease Inhibitor Cocktail, EDTA free, Roche) and pre-washed glutathione beads at 4°C for 1 hr. The beads were washed three times and eluted with 1x SDS loading buffer.

## Immunoprecipitation

Mixed-stage worms were cultured at 20°C on five 100-mm HG plates seeded with OP50, harvested, and washed with M9 buffer to yield 200–500 µl of packed worms. Then, 200 µl of packed worms was mixed with 200 µl of 2x lysis buffer A, B, or C (see below) and 800 µl of 0.5-mm diameter glass beads. The mixture was lysed using FastPrep-24 (MP Biomedicals) at 6.5 m/s, 20 s/pulse × 3 pulses with 5-min intervals on ice. Worm lysates were cleared by centrifugation at 13,000 rpm for 30 min. For anti-GFP and anti-FLAG IPs, the supernatant was incubated with GBP beads or anti-FLAG M2 beads for 1–2 hr. For the phospho-DAF-16 IP, the supernatant was incubated with 10 µl of the antibody specific for DAF-16(pT240) or DAF-16(pS298) and 30 µl of pre-washed protein A and protein G beads (mixed at 1:1) for 2 hr. After incubation with the lysates, the beads were washed 2–3 times with 1x lysis buffer, 5 min each time, and boiled in 2x SDS loading buffer for western blot analysis. All steps, from making the lysates to eluting the beads, were performed at 4°C.

## Lysis buffers

1x lysis buffer A (for IP of DAF-16 and TAX-6): 20 mM Tris pH 8.0, 1% NP-40, 10% glycerol, 2 mM EDTA, 1x PIC (Protease Inhibitor Cocktail, EDTA free; Roche).

1x lysis buffer B (for IP of DAF-16 and UNC-43): 50 mM Tris, pH 8.0, 150 mM NaCl, 0.1% NP-40, 10% glycerol, 1x PIC, 1x PhosSTOP (Roche).

1x lysis buffer C (for IP of phospho-DAF-16): 50 mM Tris, pH 7.6, 150 mM NaCl, 0.1% SDS, 0.25% sodium deoxycholate, 1% NP-40, 2 mM EDTA, 1x PIC, 1x PhosSTOP.

Immunoprecipitation of FLAG-FOXO3 from HEK293T cells was performed as described (*Xie et al., 2012*).

## In vitro kinase assay

2 µg of purified recombinant GST-UNC-43 (or 5 µg of purified active human GST-CAMKIIA) was incubated with 2 µg of CAM and 2–5 µg of purified 6xHis- or GST-tagged DAF-16 (or GST-FOXO3) in the kinase reaction buffer (50 mM PIPES, pH 7.0, 10 mM $MgCl_2$, 4 mM $CaCl_2$, 100 µM ATP, 1x PIC) in the presence or absence of [$^{32}$P]-γ-ATP at 30°C for 15 min. Proteins were resolved by SDS PAGE and analyzed by autoradiography or WB with anti-DAF-16(pT240) or anti-DAF-16(pS286) antibodies.

## In vitro phosphatase assay

The kinase reactions, performed as described above, were shifted to 60°C for 10 min to inactivate UNC-43 or CAMKIIA. Then the samples were incubated or not with 5 µg of TAX-6•GST-CNB-1 or Calcineurin in phosphatase buffer (50 mM Tris, pH 7.5, 100 mM NaCl, 6 mM $MgCl_2$, 1 mM $CaCl_2$, 1x PIC) in the presence of 2 µg CAM at 30°C for 30 min. The sample were resolved by SDS-PAGE and analyzed by autoradiography or by WB with antibodies specific for DAF-16(pT240), DAF-16(pS286), or mFOXO3(S252)/hFOXO3(S253).

## Daf-16 localization

Worms were cultured at 20°C on NGM plates from eggs laid within a 4-hr period and imaged at the L4 stage unless otherwise indicated. As soon as worms were removed from incubation, they were mounted on slides and imaged immediately. For the starvation challenge, approximately 100 L4 worms were placed on an empty plate and kept at 20°C for 20 hr before imaging. For the heat stress challenge, well-fed L4 worms on bacterial food were shifted to 28°C for 2 hr. Parallel samples were maintained at 20°C on food as controls. All GFP and DIC images were taken using a Zeiss Axio Imager M1 microscope at 400-fold magnification.

## Mass spectrometry analysis

Coomassie blue-stained bands corresponding to in vitro phosphorylated DAF-16 or FOXO3 fragments, or FLAG-FOXO3 immunoprecipetated from 293T cells were destained and in-gel digested with trypsin. LC-MS/MS analyses of the resulting peptides were carried out on an LTQ-Orbitrap (ThermoFisher Scientific) or LTQ-Orbitrap Velos (ThermoFisher Scientific) as described before with slight modifications (*Li et al., 2011*). CID MS2 with neutral-loss triggered MS3 spectra were collected on LTQ-Orbitrap and HCD MS2 spectra were collected on LTQ-Orbitrap Velos. The MS2 or MS3 data were searched against the *C. elegans* protein database (for DAF-16) or the NCBI mouse protein database (for FOXO3). The pLabel software was used for spectral labeling (*Yang et al., 2012*).

## Mammalian cell culture, transfection, and luciferase assay

HEK293T cells were maintained in DMEM medium supplemented with 10% fetal bovine serum (FBS) at 5% $CO_2$. Transfection was carried out using Lipofectamine 2000 (Invitrogen), and the luciferase assays were carried out using the Dual-Luciferase assay kit (Promega).

## Acknowledgements

The authors wish to express sincere gratitude to Drs Thomas Johnson (University of Colorado, United States) for kindly providing the $P_{daf-16}$::*daf-16*::*gfp* construct, Yamei Wang (Xiamen University, China) for providing the 14-3-3 antibody, and Joohong Ahn (Hanyang University, Korea) for the GST fusion constructs of TAX-6 and CNB-1. Some strains were provided by the CGC, which is funded by NIH Office of Research Infrastructure Programs (P40 OD010440). We thank Dr Li-Lin Du for sequence analysis of the FOXO homologs, and Dr Xiaodong Wang for critical reading of the manuscript.

## Additional information

### Funding

| Funder | Grant reference number | Author |
|---|---|---|
| The Ministry of Science and Technology of China | 863-2007AA02Z1A7, 973-2010CB835203 | Meng-Qiu Dong |
| The Ministry of Science and Technology of China | 973-2009CB918704, 973-2012CB910701 | Zengqiang Yuan |
| National Science Foundation of China | 30870792, 81030025, 81125010 | Zengqiang Yuan |
| The municipal government of Beijing | | Meng-Qiu Dong |

The funders had no role in study design, data collection and interpretation, or the decision to submit the work for publication.

### Author contributions

LT, Conception and design, Acquisition of data, Analysis and interpretation of data, Drafting or revising the article; QX, Acquisition of data, Analysis and interpretation of data; Y-HD, Acquisition of data, Contributed unpublished essential data or reagents; S-TL, SP, Y-PZ, DT, Acquisition of data; ZY, Analysis and interpretation of data, Drafting or revising the article; M-QD, Conception and design, Analysis and interpretation of data, Drafting or revising the article

## Additional files

### Supplementary files

• Supplementary file 1. (**A**) *C. elegans* strains used. (**B**) Oligos for genotyping.

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
