## [Decision Letter]

Thank you for choosing to send your work entitled “CAMKII and Calcineurin Regulate the Lifespan of *C. elegans* through the FOXO Transcription Factor DAF-16” for consideration at *eLife*. Your article has been favorably evaluated by a Senior editor and 3 reviewers, one of whom is a member of our Board of Reviewing Editors.

The Reviewing editor and the other reviewers discussed their comments before we reached this decision, and the Reviewing editor has assembled the following comments to help you prepare a revised submission.

This paper is of substantial interest based on data revealing novel interactions between *daf-16* in *C. elegans* and the orthologs of calcineurin and CamKII, powerful regulators in the mammalian immune system as well as metabolism. The data strongly support the authors’ hypothesis that *daf-16* function is positively regulated by phosphorylation by the latter and negatively regulated by dephosphorylation of the former. Furthermore, stress responses appear to be mediated by this system in controlling life span in worms. However, recent data published by Ozcan et al (cited in the present paper but not discussed in sufficient detail) have revealed phosphorylation of FOXO by CamKII, nuclear translocation, and phospho sites that are distinct from AKT sites, so these findings are not completely novel. Nonetheless, this paper extends our information substantially and the impact would be even further strengthened with extension of the findings into a mammalian system. Major concerns are noted below:

1) All the data presented related to phosphorylation/dephosphorylation have been produced through the use of expressed tagged constructs. None of the conclusions are supported by isolations of the endogenous proteins. Although the bulk of the claims based on biochemical analysis are well supported by genetic studies, direct evidence of interactions and covalent modification of endogenous proteins would strengthen the paper. It is suggested, but not required, that the authors consider using antibodies against the endogenous proteins to confirm their data.

2) The T240 site does not appear to be regulated, although again these relevant data are from expressed tagged proteins. There are no genetic studies for this site provided, which would strengthen the concept that this site is not involved.

3) The conclusions of the paper would be substantively extended and the impact strengthened if the authors could provide evidence that this site of S286 of regulation of FOXO protein in mouse and human were operational. Good reagents are available for such studies, which would be easy to do in terms of both demonstrating phosphorylation/dephosphorylation as well as nuclear/cytoplasmic localization. In this regard, even though S286 is conserved in FOXO3, S303 in FOXO1, the equivalent of S299 in FOXO3, was not identified as a site whose phosphorylation was increased by CaMKIIg by Ozcan et al., and most of the reported sites were in Ser-Pro motifs, possibly phosphorylated as a result of p38 activation.

4) The sequence around DAF-16 S286(SIQTISHDLYD) is not that of a typical site for phosphorylation by CaMKII, which is a basophilic kinase, and is not predicted by Scansite to be a CaMKII site, even at low stringency, despite the authors’ in vitro evidence that CaMKII can directly phosphorylate S286. All the mapped CaMKII sites listed in PhosphoSitePlus that have a D at +2 also have an Arg or Lys at −3, which is not the case for S286. This deserves discussion and raises the issue of whether CaMKII directly phosphorylates DAF-16 in vivo.

5) Given the focus of the field to date on AKT regulation of FOXOs, the effect of the S286 phos status on the AKT site is very interesting. The submission goes some way to determining dominance between these two regulatory mechanisms (Figure 7), but it would benefit from going further in terms of the effects on aging. Does a S286A mutation or *unc-43* null inhibit the longevity effects of DAF-2 or AKT loss of function: i.e., is phos of S286 necessary for lifespan extension by FOXO, rather than sufficient?

6) How does TAX-6/CNB-1 bind DAF-16 (i.e., what are the interaction domains in the two proteins?)?

7) How does phosphorylation of S286 promote nuclear import of DAF-16 (e.g., does it reduce 14-3-3 binding?)?

---

## [Author Response]

*1) All the data presented related to phosphorylation/dephosphorylation have been produced through the use of expressed tagged constructs. None of the conclusions are supported by isolations of the endogenous proteins. Although the bulk of the claims based on biochemical analysis are well supported by genetic studies, direct evidence of interactions and covalent modification of endogenous proteins would strengthen the paper. It is suggested, but not required, that the authors consider using antibodies against the endogenous proteins to confirm their data*.

We agree with the reviewers wholeheartedly. We have tried from an early phase of this research to detect interactions and modifications of endogenous proteins. However, none of the antibodies we made (three for DAF-16, two for TAX-6, and two for UNC-43) are good enough for IP or WB of the endogenous proteins. We have a tiny bit of TAX-6 antiserum from Dr. Joohong Ahnn that can detect endogenous TAX-6, but it doesn’t work for IP. Using this antibody we did not see endogenous TAX-6 in the DAF-16::GFP IP.

*2) The T240 site does not appear to be regulated, although again these relevant data are from expressed tagged proteins. There are no genetic studies for this site provided, which would strengthen the concept that this site is not involved*.

In the revised manuscript we provide genetic results further bolstering the conclusion that T240 is not phosphorylated by UNC-43. This is shown in Figure 6—figure supplement 1. If T240 phosphorylation mediates the effect of UNC-43, the T240A mutation should abolish DAF-16 nuclear accumulation. However, DAF-16(T240A) accumulated in the nucleus, even in an essentially WT background. This is consistent with T240 being an AKT phosphorylation site, not an UNC-43 phosphorylation site. The phenotype of DAF-16(T240A-S286A) is complex, but mostly mimicking a strong loss-of-AKT phenotype. This result again agrees with T240 being an AKT phosphorylation site and echoes with the other data shown in Figure 8, suggesting that insulin signaling overpowers UNC-43 signaling when in conflict. Lastly, DAF-16(T240A) has no effect on lifespan (below in Author response image 1), resembling DAF-16(4AM), in which four AKT consensus phosphorylation sites are all mutated to alanine ([32] Nat. Gen. and Lee et al., 2001 Curr. Biol.).

*3) The conclusions of the paper would be substantively extended and the impact strengthened if the authors could provide evidence that this site of S286 of regulation of FOXO protein in mouse and human were operational. Good reagents are available for such studies, which would be easy to do in terms of both demonstrating phosphorylation/dephosphorylation as well as nuclear/cytoplasmic localization. In this regard, even though S286 is conserved in FOXO3, S303 in FOXO1, the equivalent of S299 in FOXO3, was not identified as a site whose phosphorylation was increased by CaMKIIg by Ozcan et al., and most of the reported sites were in Ser-Pro motifs, possibly phosphorylated as a result of p38 activation*.

We have added mammalian data (Figure 9 and Figure 10) to the manuscript. Indeed, mammalian CAMKIIA and Calcineurin regulate FOXO3 phosphorylation at the same conserved site as DAF-16 S286, which is S298 in mouse FOXO3 (equivalent to S299 in human FOXO3). This regulation appears to be limited to FOXO3; phosphorylation of the equivalent site in FOXO1 by CAMKIIA was not detected. The FOXO1 results are summarized in the following paragraph, which is added to the paper under the section title “Mammalian CAMKII and Calcineurin also regulate phosphorylation of mouse FOXO3 at a conserved serine residue”:

“A recent study by Ozcan et al. shows that four non-AKT phosphorylation sites are involved in the activation of murine FOXO1 by CAMKIIγ, likely through the p38 MAP kinase, in glucagon-stimulated hepatocytes (40). Serine 300 of mouse FOXO1, the equivalent of S298 of mouse FOXO3, is not among them (S246, S295, S467, and S475). To find out whether the CAMKIIA isoform phosphorylates FOXO1 in the same way as it does FOXO3, we expressed a FLAG-tagged human FOXO1 in HEK293T cells co-transfected or not with CAMKIIA. Mass spec analysis identified three CAMKIIA-regulated phosphorylation sites equivalent to mouse FOXO1 S295, S467, and S475 (not shown). This agrees with the findings by Ozcan *et al*. and suggests that CAMKII directly phosphorylates FOXO3 but not FOXO1 in mammals.”

*4) The sequence around DAF-16 S286(SIQTISHDLYD) is not that of a typical site for phosphorylation by CaMKII, which is a basophilic kinase, and is not predicted by Scansite to be a CaMKII site, even at low stringency, despite the authors’ in vitro evidence that CaMKII can directly phosphorylate S286. All the mapped CaMKII sites listed in PhosphoSitePlus that have a D at +2 also have an Arg or Lys at −3, which is not the case for S286. This deserves discussion and raises the issue of whether CaMKII directly phosphorylates DAF-16 in vivo*.

Indeed, DAF-16 S286 is not a typical CAMKII site. However, it has been reported that CAMKII can phosphorylate Ser or Thr residue within the sequence motif S/TXD (Yamauchi, 2005, Biol Pharm Bull.). DAF-16 S286 is found within such a motif, as is mFOXO3 S298 (Figure 9). We found many examples among the mapped CAMKII sites from the PhosphoSite web page, where D at +2 is not accompanied by a K/R at -3. Below are some of them copied from PhosphoSitePlus (Author response image 2):

*5) Given the focus of the field to date on AKT regulation of FOXOs, the effect of the S286 phos status on the AKT site is very interesting. The submission goes some way to determining dominance between these two regulatory mechanisms (Figure 7), but it would benefit from going further in terms of the effects on aging. Does a S286A mutation or* unc-43 *null inhibit the longevity effects of DAF-2 or AKT loss of function: i.e., is phos of S286 necessary for lifespan extension by FOXO, rather than sufficient*?

Yes, the relationship between insulin signaling and CAMKII signaling is very interesting. We found that *unc-43(null)* significantly shortened the lifespan of *daf-2(e1370)* by 31%, suggesting that UNC-43 transmits part of the signaling from DAF-2 (Figure 7). However, the S286A mutation appeared to have no effect on lifespan extension by *daf-2* RNAi (Figure 7—figure supplement 1). We think that additional UNC-43 targets are involved in DAF-2 signaling.

*6) How does TAX-6/CNB-1 bind DAF-16 (i.e., what are the interaction domains in the two proteins?)*?

We performed pull-down assays with GST-tagged DAF-16 fragments and found that the C-terminal region of DAF-16 is important for TAX-6 binding. This is added as Figure 1. The C-terminal region harbors the regulatory site S286.

*7) How does phosphorylation of S286 promote nuclear import of DAF-16 (e.g., does it reduce 14-3-3 binding?)*?

This is a very good question but unfortunately we are not able to answer it definitively. We found that *unc-43(gf)* increased phosphorylation at S286 and decreased phosphorylation at T240 on DAF-16 (Figure 5), and this was indeed accompanied by a reduction of DAF-16-associated 14-3-3, as suspected by the reviewers (Figure 7—figure supplement 2). However, there are two possibilities for the negative correlation between S286 phosphorylation and T240 phosphorylation: AKT activity may be reduced in *unc-43(gf),* or phospho-S286 DAF-16 may be a poor substrate for AKT. Either would explain the reduction of T240 phosphorylation and 14-3-3 binding. If we had integrated strains of DAF-16(S286D)::6xHisGFP, we might have had a better chance to address this question by GFP IP and 14-3-3 WB, and finding out whether the S286D mutation reduces T240 phosphorylation. In any case, it remains to be determined whether S286 phosphorylation directly interferes with 14-3-3 binding or through reduced T240 phosphorylation.